# Brain-Inspired fMRI-to-Text Decoding via Incremental and Wrap-Up Language Modeling

**Wentao Lu**[1], **Dong Nie**[2], **Pengcheng Xue**[1], **Zheng Cui**[1], **Piji Li**[1], **Daoqiang Zhang**[1], **Xuyun Wen**[1,*]

[1]College of Artificial Intelligence, Nanjing University of Aeronautics and Astronautics, Nanjing, China
[2]ChatAlpha AI, California, USA
{luwentao,charles1231,cuizheng,pjli,dqzhang,wenxuyun}@nuaa.edu.cn
dongnie@cs.unc.edu

## Abstract

Decoding natural language text from non-invasive brain signals, such as functional magnetic resonance imaging (fMRI), remains a central challenge in brain-computer interface research. While recent advances in large language models (LLMs) have enabled open-vocabulary fMRI-to-text decoding, existing frameworks typically process the entire fMRI sequence in a single step, leading to performance degradation when handling long input sequences due to memory overload and semantic drift. To address this limitation, we propose a brain-inspired sequential fMRI-to-text decoding framework that mimics the human cognitive strategy of segmented and inductive language processing. Specifically, we divide long fMRI time series into consecutive segments aligned with optimal language comprehension length. Each segment is decoded incrementally, followed by a wrap-up mechanism that summarizes the semantic content and incorporates it as prior knowledge into subsequent decoding steps. This sequence-wise approach alleviates memory burden and ensures semantic continuity across segments. In addition, we introduce a text-guided masking strategy integrated with a masked autoencoder (MAE) framework for fMRI representation learning. This method leverages attention distributions over key semantic tokens to selectively mask the corresponding fMRI time points, and employs MAE to guide the model toward focusing on neural activity at semantically salient moments, thereby enhancing the capability of fMRI embeddings to represent textual information. Experimental results on the two datasets demonstrate that our method significantly outperforms state-of-the-art approaches, with performance gains increasing as decoding length grows. The code is available at https://github.com/WENXUYUN/CogReader.

## 1 Introduction

Language serves as a window into cognitive processes, conveying vast amounts of information through its syntactic and semantic structures [26]. Advances in non-invasive neuroimaging, such as functional magnetic resonance imaging (fMRI), have enabled researchers to measure brain activity patterns associated with language processing. Translating cognitive signals into natural language not only deepens our understanding of the neural basis of the language system, but also facilitates the development of practical brain-computer interfaces (BCIs) by leveraging insights into the decoding process [36, 23, 34].

In recent years, advances in deep learning have led to significant progress in short text generation from brain signals, such as mapping fMRI activity to semantic representations of individual words or short

---

*Corresponding author

39th Conference on Neural Information Processing Systems (NeurIPS 2025).

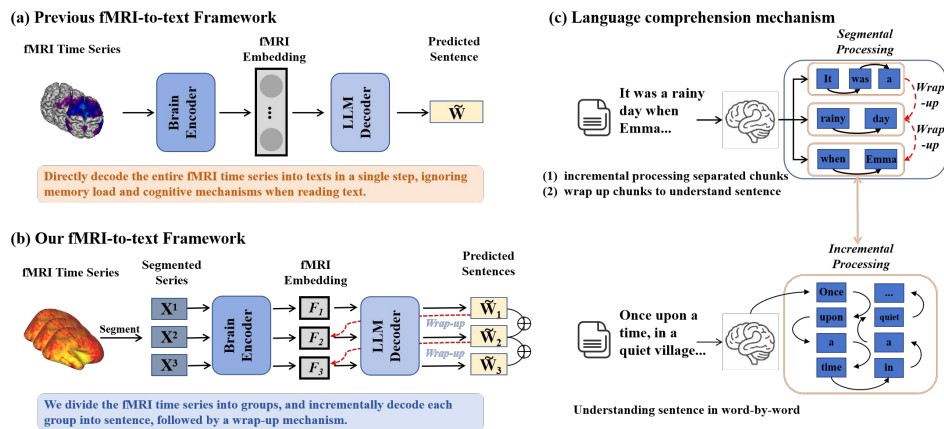

Figure 1: Comparison of fMRI-to-text decoding frameworks. (a) Existing frameworks directly decode the entire fMRI sequence corresponding to the target text in a single step. (b) Our proposed segment-based sequential decoding framework. (c) Cognitive mechanisms of human language comprehension, where incremental processing and segmental wrap-up operating in parallel.

phrases [1, 5]. However, these approaches are limited in scope, often restricted to closed vocabularies or single-word outputs, lacking the capacity to decode full natural language sentences. With the rapid development of large language models (LLM), researchers have begun to explore their application in various brain decoding tasks [8, 20, 22, 29, 9, 4]. Recent studies have shown encouraging progress in open-vocabulary fMRI-to-text generation by incorporating large language models (LLMs)[37, 3]. However, these methods still face significant challenges in decoding long sequences. As illustrated in Figure1(a), most current approaches process the entire fMRI sequence corresponding to a given text in a single step, overlooking the segmented and inductive processing strategy that the human brain adopts to manage memory load during language comprehension [14, 11]. As the length of the input increases, these approaches lead to excessive memory burden and semantic drift [25], ultimately impairing decoding performance. Unlike the traditional machine translation paradigm employed in existing methods, which enables one-to-one mapping between two independent modalities, fMRI-to-text decoding reconstructs textual content from neural activity patterns elicited during human language comprehension. Consequently, models tailored for cross-modal translation are not directly transferable to this task. Guided by this distinction, we hypothesize that a decoding framework better aligned with human language processing mechanisms would more effectively address this challenge. Therefore, it is essential to incorporate insights from human language comprehension mechanisms into the design of fMRI-to-text decoding models.

Human language understanding is neither a passive nor a strictly linear process. Instead, it emerges from a complex interplay between incremental processing and segmental integration [30], as shown in Figure 1(c). Incremental processing enables the brain to construct semantic representations in real time, interpreting linguistic input on a word-by-word basis. While this allows for immediate comprehension, it imposes a heavy load on working memory and becomes less effective for complex or long-form text. In contrast, segmental integration provides a complementary mechanism, wherein the brain periodically aggregates information across semantically coherent segments. This wrap-up process facilitates semantic consolidation and disambiguation at key structural boundaries, thereby reducing cognitive load and enhancing comprehension accuracy. Despite growing evidence supporting the importance of these dual mechanisms, they remain largely underexplored in existing fMRI-to-text decoding frameworks.

Inspired by human cognitive mechanisms for language processing, we propose a novel fMRI-to-text decoding framework that combines incremental processing with a wrap-up-based semantic integration strategy, named as **CogReader**. As shown in Figure 1 (b), we first divide the continuous fMRI time series into multiple sequential segments. For each segment, the model performs incremental decoding, generating the corresponding text word by word in real time. We also design a wrap-up integration module that summarizes the decoding results of the current segment into a semantic representation. This representation is then passed as prior knowledge to guide the decoding of the next segment,

enabling effective cross-segment information flow. Furthermore, to learn fMRI features containing more textual information, we introduce a text-guided masking strategy, integrated into a Masked Autoencoder (MAE)-based framework for fMRI representation learning. Our main contributions are summarized as follows:

1. Motivated by human language comprehension mechanisms, we design a new fMRI-to-text decoding framework that integrates incremental processing and wrap-up semantic integration. Our model enables real-time decoding for each segment and progressively incorporates cross-segment knowledge, offering an effective solution for decoding long-form text from neural activity.

2. We propose a text-guided masking strategy. By leveraging attention distributions over key semantic tokens, our method selectively masks corresponding fMRI time points and incorporates MAE to encourage the model to focus on neural activity at key time points to learn brain representations with more key textual information.

3. Extensive experiments demonstrate that our method significantly outperforms existing state-of-the-art approaches on standard fMRI-to-text decoding benchmarks. Moreover, the performance advantage becomes increasingly pronounced as sentence length grows, underscoring the feasibility and effectiveness of our cognitively inspired decoding framework.

## 2  Related Works

### 2.1  fMRI Representation Learning

Due to the complex spatiotemporal structure of fMRI data and the variability across subjects, learning robust and high-quality fMRI representations remains a significant challenge. In recent years, a variety of deep learning paradigms have been proposed to improve fMRI representation quality. For example, Kim et al. [15] utilized a variational autoencoder (VAE) [17] to model the distribution of fMRI signals while disentangling spatial and temporal components. Asadi et al. [2] proposed a hybrid model that combines spatial attention with temporal Transformers, to better model long-range spatiotemporal dependencies. While model architecture innovation has advanced fMRI representation learning, complementary learning paradigm improvements have emerged as another research focus. In fMRI-to-text decoding tasks, fully supervised learning has become standard for enhancing semantic richness of neural representations [1, 31]. Building on this foundation, contrastive learning frameworks [1, 3] further optimize cross-modal alignment by treating paired fMRI-text data as positive samples. However, the reliance on scarce paired datasets limits these supervised approaches. To mitigate this bottleneck, recent work integrates self-supervised pre-training paradigms that leverage abundant unlabeled data. For example, masked autoencoding (MAE)-inspired methods [12] have demonstrated effectiveness in capturing spatiotemporal features from raw fMRI signals through reconstruction-based learning [37], establishing a synergistic pipeline with task-specific supervision.

### 2.2  fMRI-to-text Decoding

Decoding natural language from non-invasive brain imaging modalities such as functional magnetic resonance imaging (fMRI) has long posed a core challenge in brain-computer interface (BCI) research. Early efforts predominantly focused on closed-vocabulary decoding, wherein brain signals were mapped to a fixed set of candidate words. For instance, Brain2Word [1] employed a classification-based approach to decode individual words from fMRI activity, while Défossez et al. [5] utilized contrastive learning to decode words and short phrases from auditory-evoked brain signals. With the advent of large-scale pretrained language models (LLMs), recent studies have pivoted toward open-vocabulary decoding, aiming to reconstruct fluent and unconstrained natural language from brain activity. For example, UniCoRN[37] treated fMRI time series as a foreign language and leveraged a BART-style translation architecture to generate continuous text. In addition, Tang et al. [31] employed a hybrid model that combines linear regression with a generative pretrained transformer (GPT) to perform similarity-based decoding. Most recently, BP-GPT [3] introduced a prompt-based decoding paradigm, where embeddings derived from fMRI sequences serve as prompts to condition large language models (e.g., GPT-2) for coherent text generation. To better align the modalities of brain signals and natural language, BP-GPT further integrates contrastive learning to align fMRI-derived and text-derived prompts, significantly improving decoding accuracy.

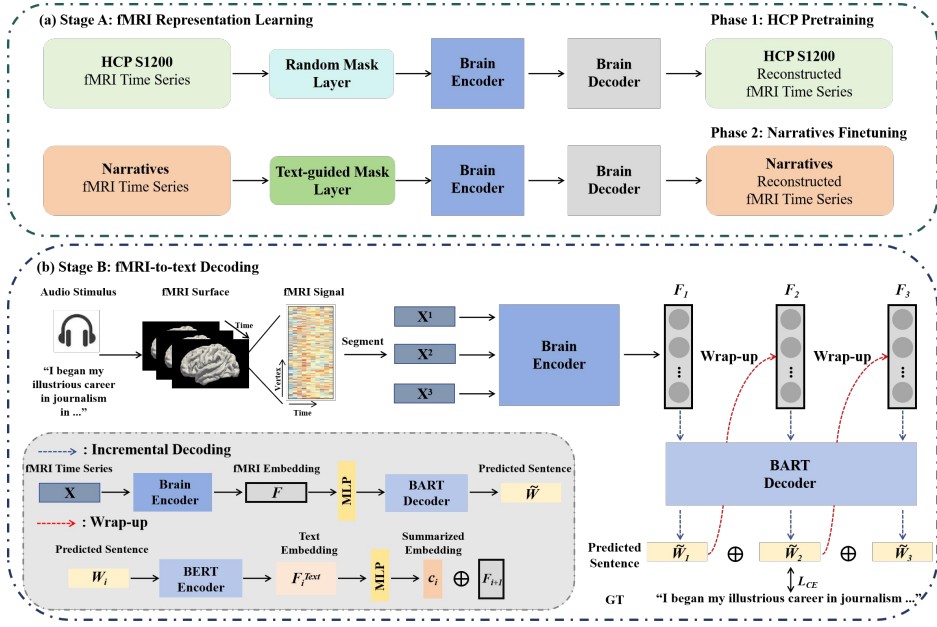

Figure 2: Framework of **CogReader**, comprising two main components: (A) fMRI representation learning and (B) fMRI-to-text decoding.

## 3    Method

In this section, we introduce the overall structure of the proposed fMRI-to-text decoding method, i.e., **CogReader**. As illustrated in Figure 2, CogReader consists of two main components: (A) fMRI representation learning and (B) fMRI-to-text decoding. In stage A, we employ a text-guided masking strategy within the MAE framework to train the brain encoder in a self-supervised manner. In stage B, a brian-inspired decoding framework is applied to generate natural language descriptions from the learned fMRI representations. Specifically, given an fMRI signal $X$ with $T$ time frames (each time frame with TR seconds), denoted as $X = \{x_1, x_2, ...x_T\}$, our goal is to decode the corresponding natural language sequence presented during scanning, represented as a sequence of words $W = \{w_1, w_2, ...w_n\}$, where each token $w_i \in V$, and $V$ denotes an open vocabulary. The implementation details of each component are described in the following subsections.

### 3.1    fMRI Representation Learning

Given the limited size of currently available fMRI-text paired datasets, it is challenging to jointly optimize both the fMRI representation learning module and the decoding module. We thus introduce a self-supervised MAE-based pretraining task (i.e., fMRI reconstruction task) prior to the decoding stage to obtain a robust brain encoder for fMRI representation learning. However, existing MAE-based approaches for fMRI representation learning suffer from a critical limitation: they fail to account for the varying semantic importance of different grouped text corresponding to different time frames. To address the above issues, we adopt a two-stage training framework for fMRI representation learning and introduce a text-guided masking strategy within the MAE to enhance the stability of the learned fMRI embeddings and improve their ability to capture the semantic information of the corresponding text.

**Two-Stage Training Strategy:** To address the data scarcity in fMRI-text paired datasets, we adopt a two-stage training process consisting of pretraining and fine-tuning, as illustrated in Figure 2(a). In the first phase, we pretrain the model on a large-scale public fMRI dataset from the Human Connectome Project (HCP), using a random masking strategy to guide the reconstruction of missing signals (mask ratio = 75%). This encourages the model to learn spatial coherence and temporal dynamics across brain regions, allowing the encoder and decoder to capture generalizable neural patterns and providing well-initialized parameters for downstream task. In the second phase, we

finetuned the HCP pretrained model on the target fMRI-text paired dataset (Narratives). Unlike conventional MAE frameworks, we introduce a text-guided masking strategy in this stage to replace the random masking approach. Through the two-phase training paradigm, we incorporate large-scale fMRI data, thereby enhancing the general representation capability of the brain encoder.

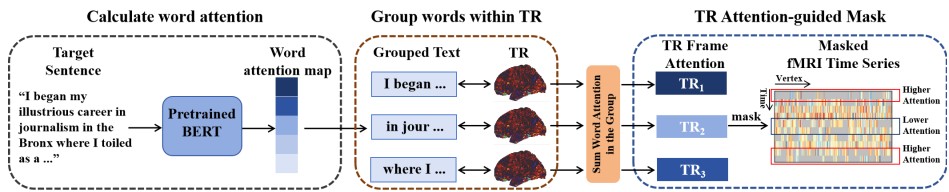

Figure 3: Illustration of text-guided masking strategy.

**Text-Guided Masking:** To enhance the semantic relevance of learned fMRI representations, we propose a text-guided masking strategy that directs the model's attention toward fMRI signals associated with highly informative semantic content, thereby improving the semantic quality of the learned representations. Given a paired input consisting of an fMRI time series $X = \{x_1, x_2, ..., x_T\}$, its corresponding stimulus text $W = \{w_1, w_2, ..., w_n\}$, we first compute the semantic importance of each word in the text using a pre-trained BERT model. Specifically, for each word $w_i$, we calculate its attention-based relevance score:

$$A_{w_i} = \frac{1}{n} \sum_{j=1}^{n} \frac{exp(\text{score}(w_j, w_i))}{\sum_{k=1}^{n} exp(\text{score}(w_j, w_k))} \tag{1}$$

where $\text{score}(w_i, w_j) = Q_i \cdot K_j^T$ represents the attention score between word $w_i$ and word $w_j$, and $Q_i$ and $K_j$ are the query and key vectors from the BERT self-attention layer. Since BERT uses multi-head attention, we average the attention scores across all heads and apply normalization to obtain the final importance score $\overline{A}_{w_i}$ for each word $w_i$:

$$\overline{A}_{w_i} = \frac{\frac{1}{H} \sum_{h=1}^{H} A_{w_i}^h}{\sum_{i=1}^{n} \frac{1}{H} \sum_{h=1}^{H} A_{w_i}^h} \tag{2}$$

where $A_{w_i}^h$ donates the attention score of word $w_i$ from head $h$, and $H$ is the total number of attention heads. To align word-level attention scores with fMRI frames (TRs), we associate each fMRI time point $x_t$ with a group of words according to fMRI-text pairs. The attention score for each TR is then computed as the sum of its associated word importance: $A_{x_t} = \sum_{w_i \in x_t} \overline{A}_{w_i}$. Using the attention vector $A_X = \{A_{x_1}, A_{x_2}, \ldots, A_{x_T}\}$, we apply a differentiable masking strategy across fMRI frames to encourage the model to focus more on neural activity during key moments. Specifically, for the top 40% of fMRI frames with the highest attention scores, we randomly mask 75% of the vertex signals. For the remaining 60% of fMRI frames with lower importance, we randomly mask only 25% of the vertex signals. This strategy encourages the model to focus more heavily on reconstructing fMRI signals from semantically rich time points, improving the quality of the learned representations and ultimately boosting downstream decoding performance.

**Autoencoder:** We employ a transformer-based autoencoder to capture the overall fMRI representation [6]. The brain encoder consists of a spatial module and a temporal module. The spatial module is designed to capture the spatial structural relationships among cortical vertices, while the temporal module models the dynamic changes across time. Both modules are composed of multiple stacked Transformer blocks, each receiving positional embeddings corresponding to either spatial or temporal locations. Specifically, given the input fMRI time series $X = \{x_1, x_2, ..., x_T\}$, the spatial module first projects it through a linear layer, followed by a stack of 8 Transformer blocks, producing a feature matrix $F_t^s \in \mathbb{R}^{N \times d_{\text{spat}}}$, where $N$ is the number of vertices and $d_{\text{spat}}$ is the dimension of the spatial module. A global average pooling layer then aggregates these vertex-level features into a single vector $s_t \in \mathbb{R}^{d_{\text{spat}}}$ representing the spatial embedding at time $t$. The temporal module then takes the sequence of spatial embeddings $\{s_t\}_{t=1}^{T}$ and encodes temporal dependencies via 8 Transformer layers, producing a latent sequence $\{h_t\}_{t=1}^{T}$, $h_t \in \mathbb{R}^{d_{\text{temp}}}$. The brain decoder takes $\{h_t\}_{t=1}^{T}$ as input and adopts 4 transformer blocks and a linear layer to reconstruct the fMRI time series $\tilde{X} = \{\tilde{x_1}, \tilde{x_2}, ..., \tilde{x_T}\}$.

**Reconstruction Loss:** Considering that fMRI signals are time series with strong temporal dependencies, our reconstruction loss is designed to account for both the masked and unmasked portions of the sequence. Specifically, given the original fMRI signal $X$ and the reconstructed fMRI signal $\tilde{X}$, the reconstruction loss is formatted as $L_{\text{recon}} = \text{MSE}(X, \tilde{X})$.

## 3.2 fMRI-to-Text Decoding

Current fMRI-to-text decoding frameworks typically generate complete text segments directly from the entire fMRI sequence, neglecting the segmented and inductive processing strategy adopted by the human brain to manage memory load during language comprehension. This often results in performance degradation when decoding long text sequences. To address this issue, we propose a brain-inspired sequence-by-sequence fMRI-to-text decoding framework that integrates incremental decoding with a semantic wrap-up mechanism, as illustrated in Figure 2(b). Specifically, we divide the long fMRI sequence into consecutive segments, each aligned with the optimal length for human language processing (discussed in section 4.4). Within each segment, we perform incremental decoding to generate partial textual outputs. After decoding each segment, a wrap-up mechanism summarizes its semantic content, which is then incorporated as prior knowledge into the subsequent segment's decoding process. This sequence-wise decoding strategy not only mitigates the memory burden and performance drop associated with long-text decoding but also ensures semantic continuity across successive segments.

**Incremental Decoding within fMRI Segments:** Given an input fMRI time series $X = \{x_1, x_2, \ldots, x_T\}$, we first divide it into consecutive segments of equal length, where each segment contains $N_s$ fMRI frames. The segmented sequence is denoted as $\overline{X} = \{X^1, X^2, \ldots, X^K\}$, where $K = \frac{T}{N_s}$. The optimal $N_s$ is discussed in Section 4.2. For each fMRI segment, we first employ a brain encoder to extract its corresponding representation, which is then fed into a well-established brain-to-text decoder to directly generate the associated text. This process is referred to as incremental decoding, resembling the human brain's real-time comprehension of incoming language input. In this study, we adopt the BART model as the fMRI-to-text decoder, due to its well performance on language understanding tasks and its suitability for sequence-to-sequence reconstruction [33, 19]. This aligns well with our task structure, where the fMRI representations are translated into corresponding textual sequences [37]. Specifically, given the fMRI segment $X^i$, we first use brain encoder to extract its fMRI representation $F_i \in \mathbb{R}^{N_s \times d_{fMRI}}$, where $d_{fMRI}$ is the number of feature dimensions. Next, a linear projection layer is used to map $F_i$ into the embedding space of the BART decoder, resulting in $F_i^{\text{BART}} \in \mathbb{R}^{N_s \times d_{\text{BART}}}$, where $d_{\text{BART}}$ represents the dimensionality of the decoder's embedding space. Finally, the projected embedding $F_i^{\text{BART}}$ is fed into the BART model to generate the predicted text $\tilde{W}_i$ corresponding to the current fMRI segment.

**Semantic Wrap-Up across fMRI Segments:** To address the potential issue of semantic discontinuity across segments in the final decoded text, we incorporate a wrap-up mechanism into the sequential decoding framework, inspired by the human brain's cognitive strategy for integrating information during language comprehension. Specifically, after obtaining the decoding text $\tilde{W}_i$ from the fMRI segment $X^i$ via incremental decoding, we employ a pretrained BERT [7] model to extract its contextualized embedding representation, denoted as $F_i^{\text{text}}$. This embedding is then passed through an MLP layer $P_\theta$ to derive a semantic summary vector:

$$c_i = P_\theta(F_i^{text}) \tag{3}$$

This process is designed to simulate the inductive summarization mechanism observed in human reading. During the decoding of the next fMRI segment $X^{i+1}$, we incorporate the summarized embedding $c_i$ into its corresponding representation vector $F_{i+1}$, guiding the decoding process for $X^{i+1}$. By incorporating the summarized semantic knowledge from the previous text segment into the decoding of the subsequent segment, the model enhances semantic continuity across successive segments. The dimensionality of the MLP $P_\theta$ is discussed in detail in Section 4.2.

**Decoding Loss:** Through incremental decoding and semantic wrap-up, we decode the complete text $\tilde{W} = \left\{ \tilde{W}_1, \tilde{W}_2, \ldots \tilde{W}_K \right\}$ from fMRI segments $\overline{X} = \{X^1, X^2, \ldots, X^K\}$ in a sequence-by-sequence manner. The objective function for the decoding stage is defined as the cross-entropy loss between

the generated text $\tilde{W}$ and the corresponding ground-truth text $W$, which is formatted as

$$L_{decoding} = \sum_{i=1}^{K} CE\left(W_i, \tilde{W}_i\right) \tag{4}$$

Here, $CE(\overline{w}_i, w_i) = -\sum_{i=1}^{N_t^i} w_i \log(\tilde{w}_i)$, where $N_t^i$ is the number of words in text squence $W_i$.

## 4 Experiments

### 4.1 Experimental Setup

**Datasets** This study employs three neuroimaging datasets: HCP S1200 [32], Narratives [24] and Huth dataset [18]. The Human Connectome Project's HCP S1200 dataset provides extensive fMRI data from 1,206 healthy young adults across seven cognitive domains. We primarily use this dataset to pretrain the brain encoder for fMRI representation learning, addressing the scarcity of fMRI-text paired data and enhancing the encoder's generalization. The Narratives dataset, a paired fMRI-text benchmark, contains fMRI recordings from 345 participants during naturalistic auditory comprehension of 27 real-world narrative stories, totaling approximately 6.4 days of functional imaging data. The Huth dataset comprises fMRI data from 8 subjects recorded while they passively listened to naturally spoken English stories, and the stories were sourced from The Month and New York Times Modern Love podcasts. Narratives dataset and Huth Dataset are used for the decoding task. All fMRI data from these datasets were preprocessed [10] and projected onto the cortical surface using the standardized preprocessing pipelines provided by each source.

**Implementaion Details** Our model is built using the PyTorch framework [28] and the Huggingface Transformers package [35]. All models utilize the Adam optimizer [16], with a warmup strategy. All experiments are conducted on CUDA 12.2 and the computer with NVIDIA GeForce RTX 3090 GPU. Additional implementation details can be found in the Appendix.

**Evaluation Metrics** To comprehensively evaluate decoding performance, we adopt three text generation metrics: BLEU-N [27], ROUGE-1 [21], and BERTScore [39]. Among them, BLEU-N and ROUGE-1 assess word-level overlaps between the generated and reference texts, while BERTScore evaluates semantic similarity based on contextual embeddings. Specifically, we report BLEU-1 to BLEU-4 scores for BLEU-N; ROUGE-F, ROUGE-P, and ROUGE-R scores for ROUGE-1; and BERTScore-F, BERTScore-P, and BERTScore-R for BERTScore.

### 4.2 Parameter Settings

This section discusses the optimal configuration of two key parameters for the fMRI-to-image decoding task. The evaluation is conducted on the Narrative dataset using BLEU-1, ROUGE-R, and BERTScore-R as performance metrics.

**Segment Length $N_s$:** To determine the optimal segment length $N_S$, we vary it from 10 to 70 in steps of 10. For each value of $N_s$, we train and test the CogReader framework accordingly. As shown in Figure 4, all metrics exhibit a trend of first increasing and then decreasing with longer segment lengths, reaching peak performance at $N_S = 20$. Therefore, we set $N_S = 20$ for all subsequent experiments.

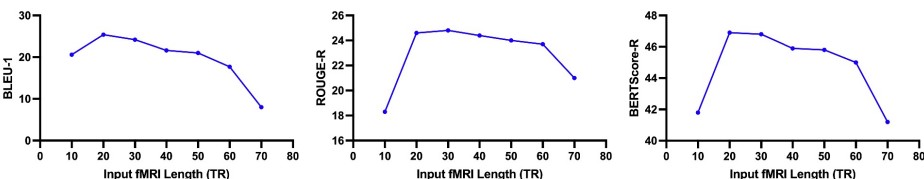

Figure 4: Parameter settings for segment length $N_s$ in incremental decoding.

**Dimensionality of the MLP:** For the MLP dimensionality, we evaluate five configurations: 0, 32, 64, 128, and 256. The corresponding performance of CogReader under each setting is shown in Figure 5. Taking into account both word-level and semantic-level evaluation metrics, we ultimately set the

MLP dimension to 128, as this configuration demonstrates consistently good and stable performance across all metrics. In comparison, other configurations perform well on at most a single metric.

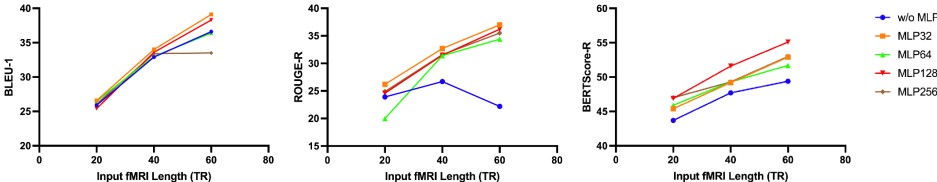

Figure 5: Parameter settings for dimensionality of the MLP in semantic wrap-up.

## 4.3 Comparison with State-of-the-art methods

We compared our proposed fMRI-to-text method with four state-of-the-art (SOTA) approaches: UniCoRN [37], EEG-Text [33], BP-GPT [3] and PREDFT [38]. Similar to our method, UniCoRN adopts a two-stage decoding framework consisting of fMRI representation learning followed by fMRI-to-text decoding. EEG-Text was originally designed for EEG signal generation tasks. We retrained its encoder using fMRI data to adapt it for fMRI decoding. BP-GPT is a prompt-based decoding method that guides text generation by aligning fMRI representations with text embeddings via contrastive learning. PREDFT is an end-to-end fMRI-to-text decoding model that jointly models neural decoding and brain predictive coding. In our experiments, we conducted comparisons under three different text decoding lengths, corresponding to fMRI time series of 20 TRs, 40 TRs, and 60 TRs. The comparative results in Narratives are given in Tables 1 and 2. The results in Huth dataset can be found in the Appendix A.3. Further qualitative analyses of decoded text cases are included in Appendix A.4.

Table 1: Comparison of our method and SOTA methods under different text decoding lengths on the Narrative dataset.

| Length | Method | BLEU-N(%) | | | | ROUGE-1(%) | | | BERTScore(%) | | |
|---|---|---|---|---|---|---|---|---|---|---|---|
| | | BLEU-1 | BLEU-2 | BLEU-3 | BLEU-4 | ROUGE-F | ROUGE-P | ROUGE-R | BERTScore-F | BERTScore-P | BERTScore-R |
| 20TR | UniCoRN | 22.9 | 2.5 | 0.3 | 0 | 20.3 | 19.6 | 21 | 43.9 | 44.2 | 42.8 |
| | EEG-Text | 24.6 | 9.3 | 4.4 | 1.9 | 21.9 | 21.1 | 23.4 | 44.6 | 43.9 | 45.4 |
| | BP-GPT | 21.6 | 3.8 | 2.5 | 1.7 | 21.6 | 20.9 | 23.4 | 44.1 | 42.1 | 46.3 |
| | PREDFT | 24.3 | 4.2 | 0.7 | 0.1 | 20.1 | 22.3 | 18.3 | 45.9 | 45.5 | 46.7 |
| | CogReader(ours) | **25.4** | **10.5** | **4.7** | **2.6** | **23.4** | **22.6** | **24.6** | **46.3** | **45.7** | **46.9** |
| 40TR | UniCoRN | 19.1 | 2.3 | 0.5 | 0.1 | 17.8 | 18.2 | 17.6 | 43.8 | 44.8 | 42 |
| | EEG-Text | 20.1 | 7.3 | 3 | 1.3 | 24.4 | 25.1 | 24.7 | 45.4 | 45.8 | 45.5 |
| | BP-GPT | 19.9 | 3.6 | 2.3 | 1.5 | 21.1 | 19.3 | 22.9 | 42.6 | 39.4 | 46.1 |
| | PREDFT | 25.9 | 4.8 | 1.4 | 0.4 | 21.1 | 24.8 | 18.6 | 46.3 | 46.2 | 46.8 |
| | CogReader(ours) | **31.2** | **15.3** | **10.3** | **8.2** | **29.6** | **28.7** | **30.4** | **50** | **49.3** | **51.1** |
| 60TR | UniCoRN | 18 | 1.7 | 0.2 | 0.4 | 16.5 | 15.9 | 17 | 43.2 | 43.7 | 42.7 |
| | EEG-Text | 22.1 | 8.2 | 3.4 | 1.6 | 28.1 | 29.4 | 28.1 | 47.7 | 47.8 | 47.7 |
| | BP-GPT | 19.3 | 3.4 | 1.3 | 0.6 | 19.4 | 19.6 | 19.3 | 41.6 | 38.2 | 45.3 |
| | PREDFT | 26.4 | 6.1 | 1.9 | 0.6 | 28.1 | 25.5 | 20.5 | 48.1 | 47.7 | 48.5 |
| | CogReader(ours) | **36.2** | **20.4** | **14.7** | **12.1** | **36.2** | **35.6** | **37.2** | **53.5** | **52.6** | **54.5** |

**Quantitative Comparison** As shown in Table 1, our method consistently outperforms all SOTA methods across all decoding lengths and evaluation metrics, demonstrating the overall effectiveness of our proposed brain-inspired framework. From the perspective of word-level metrics (BLEU-N and ROUGE-1), the performance of SOTA methods degrades as the length of the decoded text increases, whereas our method exhibits an upward trend. In terms of semantic-level evaluation (BERTScore), the SOTA methods show relatively stable performance, while our method continues to improve with longer decoding sequences. Similar results are also observed on the Huth dataset (Appendix A.3). Under the 60TR time window, our method achieves significantly better text decoding performance than other approaches.

**Decoding Text Comparison** As shown in Table 2, our method consistently outperforms others in both semantic information extraction and narrative consistency. For instance, our model accurately identifies and generates key phrases such as "Pie Man", "Ego sum non an bestia", "scene", "said", and "fled", while producing more coherent and contextually appropriate sentences. In contrast, other methods exhibit greater deviations from the target content. These results suggest that the advantage of our approach becomes increasingly evident as the decoding length grows, highlighting the effectiveness and feasibility of the brain-inspired sequential decoding strategy we adopt.

Table 2: Cases analysis for fMRI-text decoding on Narrative dataset. Exact matches between the target and predicted sentences are indicated in **bold**, while semantic similarity is shown in *italic* font.

| Results |
|---|
| **Target**: *was* I gave **the** figure a name **I** called **him Pie Man** capital P capital M **And I** described **him as** a cape wearing masked avenger Though **in** fact **he**'d **been** capeless **And said that as he fled the scene he** clicked **his** heels **in** rakish glee **And I** gave **him** a catchphrase **in** Latin **I said that he cried out Ego sum non an bestia which** means **I am** *not* an animal **Which makes** no sense |
| **UniCoRN**: **and** me gray to **the** little **Man and** no taller than my knee, with a coat too big for **his** frame lingered by **the** gate. It's **been** nigh of years **he** to **the** then **said** a case here. **The** leather sat weathered on **the** step, its lock rusted shut, but **I** knew what **was** inside: **the** girl of **the** story, **the** one who scribbled tales of dragons **in the** margins of her notebooks, had tucked her most precious drawings there before she |
| **EEG-Text**: **and** this is where it gets really interesting **I**'*m* trying to think of a title for **the** book **and I** finally come up with **an** idea **that** sounds like something **out** of one of those old pulp stories by John Wilmot who lived **in** Waukegan Illinois but **he was** actually born **in** Westchester County New York **which** is about twenty five miles outside Chicago So **I am** going to describe my novel **as The Man** Who Walks into Times Square **and** stops before **he** can get too lost |
| **BP-GPT**: **I**'*m* a teacher **and I** teach people how to make things better by using my own talents **And** so one day **I was** walking down **the** street **and** this old lady came up to me **and** she **said** You know you sound like **an** intelligent black woman So **I** told her **that**'s exactly what **I**'*m* here to do |
| **PREDFT**: **the** a **said the** girl a **man I said the Man and** use of best **and** she me **and I man** fl of Litgo be **and the** It's flirting **and** she guy of **the** eyes thelee raised **non** crate **and** she littleiving it then to of crate **non** a best owan is best me best to |
| **Ours**: **I** think **that** you realize what happened next **Pie Man** emerged from **the** late night library drop **made his** delivery **and fled** away **crying Ego sum non an bestia** Or that'*s* what it **said in** my story **in the** newspaper next day **which** ran with photos of **him** leaving **the scene** cape flowing behind **him** doing this **And I'm** just like praying my life does*n't* flash before my eyes **and** ruins |

## 4.4 Ablation Study

In this subsection, we conduct an ablation study to evaluate the effectiveness of three key components: the HCP pretraining phase, the text-guided masking strategy, and the sequential fMRI-to-text decoding framework. For testing the masking strategy, we replace the proposed text-guided masking with a conventional random masking scheme. The experiments are conducted on decoding tasks with fMRI time series of 60 TRs on the Narrative dataset. The results are reported in Table 3. The results show a consistent improvement in performance as each module is incrementally added, validating the effectiveness of each individual component. Notably, the brain-inspired sequential decoding framework yields the most significant performance gain, further demonstrating the feasibility and impact of our proposed decoding approach.

Table 3: Ablation Study of our method

| Method | | | BLEU-N(%) | | | | ROUGE-1(%) | | | BERTScore(%) | | |
|---|---|---|---|---|---|---|---|---|---|---|---|---|
| Sequential Decoding | Pretraining | Text-guided Masking | BLEU-1 | BLEU-2 | BLEU-3 | BLEU-4 | ROUGE-F | ROUGE-P | ROUGE-R | BERTScore-F | BERTScore-P | BERTScore-R |
| ✗ | ✗ | ✗ | 17.7 | 6.5 | 2.4 | 1.1 | 29.2 | 32.9 | 24.7 | 46.7 | 47.6 | 45 |
| ✓ | ✗ | ✗ | 32.5 | 16.5 | 11.1 | 8.9 | 28.2 | 25.8 | 30.6 | 51.1 | 50.3 | 51.8 |
| ✓ | ✓ | ✗ | 34.0 | 18.1 | 12.7 | 10.2 | 34.1 | 33.7 | 35.7 | 52.3 | 51.0 | 53.7 |
| ✗ | ✓ | ✓ | 21.6 | 7.9 | 3.2 | 1.5 | 26.6 | 29.4 | 25 | 47.4 | 47.7 | 47.2 |
| ✓ | ✓ | ✓ | **36.2** | **20.4** | **14.7** | **12.1** | **36.2** | **35.6** | **37.2** | **53.5** | **52.6** | **54.5** |

## 4.5 Evaluation on fMRI Representation Learning

In this section, we evaluate the effectiveness of fMRI representation learning on the Narrative dataset. To this end, we design two experiments, including a comparison with other representation learning methods and an analysis against noise data.

**Comparison with other fMRI Representation Learning Method** Since the overall framework of UniCoRN is similar to ours in the current fMRI-to-text decoding paradigm, we take UniCoRN as a baseline and replace its fMRI representation learning module with our proposed method, while keeping the fMRI-to-text decoding strategy unchanged. This design allows us to evaluate the effectiveness of our representation learning approach. We conduct comparison experiments on fMRI time series ranging from 10 TRs to 50 TRs. The results, shown in Table 4, demonstrate that

under the same decoding strategy, replacing the representation learning component with our method consistently improves decoding performance across all sequence lengths, validating the effectiveness of the proposed representation learning framework.

Table 4: Comparison results of our fMRI representation learning method with other methods

| Length | Method | BLEU-N(%) | | | | ROUGE-1(%) | | | BERTScore(%) | | |
|---|---|---|---|---|---|---|---|---|---|---|---|
| | | BLEU-1 | BLEU-2 | BLEU-3 | BLEU-4 | ROUGE-F | ROUGE-P | ROUGE-R | BERTScore-F | BERTScore-P | BERTScore-R |
| 10TR | UniCoRN | 18.1 | 2.9 | 0.4 | 0 | 10.5 | 10.2 | 16.6 | 40.2 | 40.1 | 40.4 |
| | Ours | 20.6 | 7 | 2.8 | 1.3 | 17.1 | 16.2 | 18.3 | 41.1 | 40.5 | 41.8 |
| 20TR | UniCoRN | 22.9 | 2.5 | 0.3 | 0 | 20.3 | 19.6 | 21 | 43.9 | 44.2 | 42.8 |
| | Ours | 25.4 | 10.5 | 4.7 | 2.6 | 23.4 | 22.6 | 24.6 | 46.3 | 45.7 | 46.9 |
| 30TR | UniCoRN | 20.3 | 2.8 | 0.5 | 0.1 | 18.3 | 18.3 | 18.4 | 41.4 | 41.5 | 41.4 |
| | Ours | 24.2 | 9.1 | 3.9 | 1.8 | 25.1 | 26.2 | 24.8 | 47 | 47.1 | 46.8 |
| 40TR | UniCoRN | 19.1 | 2.3 | 0.5 | 0.1 | 17.8 | 18.2 | 17.6 | 43.8 | 44.8 | 42 |
| | Ours | 21.6 | 7.9 | 3.2 | 1.5 | 25.2 | 27 | 24.4 | 46.1 | 46.2 | 45.9 |
| 50TR | UniCoRN | 18.9 | 1.9 | 1.8 | 1.1 | 17.3 | 16.8 | 17.4 | 44.8 | 43.9 | 45.7 |
| | Ours | 21 | 7.7 | 3.2 | 1.5 | 26.1 | 29.4 | 24 | 46.5 | 47.7 | 45.8 |

**Comparison with Noise Data** Previous work [13] has shown that existing open-vocabulary brain decoding methods often yield similar performance on both real and noise data, suggesting that these approaches fail to effectively capture the semantic information encoded in brain signals and instead rely heavily on the memory capacity of the large language model (LLM) decoder. To address this concern, we evaluate the effectiveness of our proposed CogReader model using both real fMRI inputs and noise data as input. The experiment is conducted on fMRI time series with a length of 60 TRs. The experimental results are presented in Table 5. The results show that decoding performance is significantly higher when using real fMRI data compared to noise input, providing strong evidence that our brain-inspired method is capable of extracting meaningful semantic information from fMRI time series, rather than depending solely on the memorization ability of the LLM.

Table 5: Comparison results between real fMRI data and noise data

| Data | | BLEU-N(%) | | | | ROUGE-1(%) | | | BERTScore(%) | | |
|---|---|---|---|---|---|---|---|---|---|---|---|
| Train | Test | BLEU-1 | BLEU-2 | BLEU-3 | BLEU-4 | ROUGE-F | ROUGE-P | ROUGE-R | BERTScore-F | BERTScore-P | BERTScore-R |
| Noise | Noise | 27.5 | 9.4 | 4.6 | 1.8 | 25.6 | 26.1 | 25.5 | 48.2 | 48.0 | 48.4 |
| Noise | fMRI | 25.3 | 7.2 | 2.4 | 1.4 | 23.9 | 23.5 | 24.8 | 47.7 | 47.2 | 48.3 |
| fMRI | Noise | 26.8 | 7.7 | 2.7 | 1.2 | 23.9 | 23.1 | 24.9 | 47.9 | 47.4 | 48.5 |
| fMRI | fMRI | **36.2** | **20.4** | **14.7** | **12.1** | **36.2** | **35.6** | **37.2** | **53.5** | **52.6** | **54.5** |

## 5    Discussion

**Limitations and Future Work** In the current method, the segment length is determined based on optimal decoding performance in a fixed setting and cannot dynamically adjust to the complexity of different stimulus texts. This static segmentation strategy may constrain further improvements in decoding accuracy. Future work could explore content-adaptive segmentation approaches that dynamically predict segment boundaries based on narrative complexity, enabling more flexible adaptation to diverse textual inputs. In light of the relatively low temporal resolution of fMRI, where each TR corresponds to multiple words, making it harder to generate coherent and complete sentences. Future work could explore the strengths of integrating fMRI with EEG, as fMRI offers semantic representation capabilities, while EEG provides high temporal resolution, which may help improve decoding accuracy. Moreover, due to the limited availability of paired fMRI-to-text datasets, our model was evaluated on a single public dataset. We plan to validate the robustness and generalizability of our approach on multiple datasets in future studies.

**Conclusion** This work proposes a brain-inspired sequential fMRI-to-text decoding framework that mimics the human cognitive strategy of segmented and incremental language processing. This method divides long fMRI sequences into optimal-length segments, each of which is decoded incrementally. A wrap-up mechanism is employed between segments to integrate and propagate semantic information, thereby alleviating memory burden and preserving semantic coherence across the entire sequence.

## Acknowledgements

This work was supported by the National Natural Science Foundation of China (62476129) and the STI 2030-Major Projects (2022ZD0209000).

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

# A Technical Appendices and Supplementary Material

## A.1 Implementation Details

**BLEU-N Metric Implementation** We use the BLEU-N scores as the main evaluation metrics for word-level similarity. This evaluation method is summarized in the algorithm below:

---
**Algorithm 1** BLEU-N Score calculation process

---
**Require:** Predicted sentence $\tilde{W}$, Reference sentence(s) $W$, n-gram order $n$
**Ensure:** BLEU-N score list $P$
1: Initialize: $P \leftarrow$ list of n-gram precisions
2: **for** $i = 1$ to $n$ **do**
3:      $num_{pred} \leftarrow$ Count all $i$-grams in $\tilde{W}$
4:      $num_{ref} \leftarrow$ Count all $i$-grams in $W$
5:      $num_{overlap} \leftarrow \sum_{w \in \tilde{W} \cap W} \min(num_{ref}, num_{pred})$,
6:      $p_i \leftarrow \frac{num_{overlap}}{num_{pred}}$
7:      Add $p_i$ to $P$
8: **end for**

---

**ROUGE-1 Metric Implementation** We compute the ROUGE-1 precision, recall, and F1 score to evaluate unigram overlap between the predicted and reference sentences. The computation process is as follows:

---
**Algorithm 2** ROUGE-1 Score (Precision, Recall, F1) calculation process

---
**Require:** Predicted sentence $\tilde{W}$, Reference sentence $W$
**Ensure:** ROUGE-1 Recall $R$, Precision $P$, and F1 Score $F$
1: Extract all unigrams $w$ from $\tilde{W}$ and $W$
2: Count overlapping unigrams between $\tilde{W}$ and $W$
3: $num_{overlap} \leftarrow \sum_{w \in \tilde{W} \cap W} \min(\text{count}_{\tilde{W}}, \text{count}_W)$
4: $num_{ref} \leftarrow$ total number of unigrams in $W$
5: $num_{pred} \leftarrow$ total number of unigrams in $\tilde{W}$
6: $R \leftarrow \frac{num_{overlap}}{num_{ref}}$
7: $P \leftarrow \frac{num_{overlap}}{num_{pred}}$
8: **if** $R + P = 0$ **then**
9:      $F \leftarrow 0$
10: **else**
11:      $F \leftarrow \frac{2 \cdot R \cdot P}{R + P}$
12: **end if**

---

**BERTScore Metric Implementation** We choose BERTScore to evaluate the similarity between the predicted sentences and the reference sentences by computing the alignment-based similarity of contextualized token embeddings. The computation steps are as follows:

## A.2 Experimental Settings

In the fMRI Repersentation Learning Stage, during HCP pretraining phase, we split the HCP dataset into training and testing sets in a 4:1 ratio. While for the Narratives dataset, to avoid text leakage, we adopt a stimulus split approach to ensure that the train, validation and test sets use different story content, with a ratio of 60%, 20% and 20%, during Narratives pretraining stage and fMRI-to-text decoding stage. We finalized the model parameters as detailed Table 6.

During the phase 2 in fMRI Representation Stage, we adopt the proposed Text-guided masking strategy to finetuned the HCP pretrained model on the target fMRI-text paired dataset. The parameters are shown in Table 7.

---
**Algorithm 3** BERTScore (Precision, Recall, F1) Calculation
---
**Require:** Pretrained language model $M(\cdot)$, Predicted sentence $\tilde{W}$, Reference sentence $W$
**Ensure:** BERTScore Precision $P$, Recall $R$, F1 Score $F$
1: Tokenize $\tilde{W}$ and $W$ into subword tokens
2: Compute contextual embeddings: $E_{\tilde{W}} \leftarrow M(\tilde{W})$, $E_W \leftarrow M(W)$
3: Compute cosine similarity matrix $S[i,j] = \cos(E_{\tilde{W}}[i], E_W[j])$
4: $P \leftarrow \frac{1}{|\tilde{W}|} \sum_{i=1}^{|\tilde{W}|} \max_{1 \leq j \leq |W|} S[i,j]$
5: $R \leftarrow \frac{1}{|W|} \sum_{j=1}^{|W|} \max_{1 \leq i \leq |\tilde{W}|} S[i,j]$
6: **if** $P + R = 0$ **then**
7: $\quad F \leftarrow 0$
8: **else**
9: $\quad F \leftarrow \frac{2PR}{P+R}$
10: **end if**
---

Table 6: **Parameter setting in HCP Pretraining.**

| parameter | value | parameter | value | parameter | value |
|---|---|---|---|---|---|
| mask ratio | 0.75 | epochs | 20 | Temporal encoder embed dim | 64 |
| batch size | 32 | warm-up epochs | 5 | Temporal encoder depth | 8 |
| optimizer | Adam | initial LR | 1e-4 | Temporal encoder heads | 2 |
| LR scheduler | StepLR | Spatial encoder embed dim | 128 | decoder embed dim | 64 |
| step size | 5 | Spatial encoder depth | 8 | decoder depth | 4 |
| gamma | 0.5 | Spatial encoder heads | 2 | decoder heads | 2 |

Table 7: **Parameter setting in Narratives Finetuning.**

| parameter | value | parameter | value | parameter | value |
|---|---|---|---|---|---|
| high attention ratio | 0.4 | batch size | 16 | LR scheduler | StepLR |
| mask ratio for high attention | 0.75 | epochs | 40 | step size | 5 |
| low attetnion ratio | 0.6 | optimizer | Adam | gamma | 0.5 |
| mask ratio for low attention | 0.25 | initial LR | 1e-4 | warm-up epochs | 5 |

For the fMRI-to-text decoding, we use the pretrained spatial-temporal encoder from Stage A and BART as the decoder for text generation from fMRI embedding, with a pretrained BERT as the encoder for summarized embedding. The detailed parameters in this stage are shown in Table 8.

Table 8: **Parameter setting in fMRI-to-text decoding.**

| parameter | value | parameter | value | parameter | value |
|---|---|---|---|---|---|
| batch size | 16 | LR scheduler | StepLR | fMRI embed dim | 256 |
| epochs | 20 | step size | 5 | BART embed dim | 1024 |
| optimizer | Adam | gamma | 0.5 | BERT embed dim | 768 |
| initial LR | 1e-5 | warm-up epochs | 5 | | |

The equipment used in the experiment is configured as follows: AMAX Tower Workstation TS40-X3, equipped with dual Intel Xeon 4316 CPUs (2.3 GHz, 20 cores), 256 GB of DDR4 memory (32 GB modules at 3200 MHz), a 480 GB SSD for the system disk, a 3.84 TB SSD for hot data, and a 16 TB 7200 RPM SATA enterprise HDD for data storage. The system is powered by dual power supplies rated at 2000W and 1650W.

## A.3 Other Results

We evaluate our method on Huth dataset obtained during apassive natural language listening task [18]. The comparative results with four SOTA methods in 60 TR are given in Table 9 below.

Table 9: Comparison between our method and SOTA methods

| Length | Method | BLEU-N(%) | | | | ROUGE-1(%) | | | BERTScore(%) | | |
|---|---|---|---|---|---|---|---|---|---|---|---|
| | | BLEU-1 | BLEU-2 | BLEU-3 | BLEU-4 | ROUGE-F | ROUGE-P | ROUGE-R | BERTScore-F | BERTScore-P | BERTScore-R |
| 60TR | UniCoRN | 21.5 | 5.4 | 2.3 | 0.6 | 17.1 | 14.1 | 19.7 | 40.1 | 37.7 | 42.7 |
| | EEG-Text | 23.6 | 7.5 | 2.2 | 1.0 | 27.6 | 28.1 | 26.1 | 47.2 | 47.3 | 47.2 |
| | BP-GPT | 20.7 | 8.1 | 2.6 | 1.2 | 27.1 | 30.4 | 23.5 | 50.7 | 51.3 | 49.2 |
| | PREDFT | 22.6 | 6.7 | 2.5 | 1.1 | 24.6 | 23.5 | 26.6 | 45.5 | 42.8 | 48.9 |
| | CogReader(ours) | **35.2** | **18.0** | **10.2** | **7.1** | **32.2** | **34.0** | **30.9** | **53.9** | **54.5** | **53.3** |

## A.4 Text cases

Table 10 and 11 present the comparision results of representative examples in Narratives, comparing the predicted sentence and the reference sentence in different fMRI time series lengths, including 20 TRs, 40 TRs, and 60 TRs between our propoesed method CogReader and SOTA methods.

Table 10: Cases analysis for fMRI2text. Exact matches between the target and predicted sentences are indicated in **bold**, while semantic similarity is shown in *italic* font.

| fMRI Length | Results |
|---|---|
| 20TR | **Target**: ... on campus People started dressing ... and quoting him in class The Ram ran ... stories about Pie Man all ... me And toward ... and I saw I came and I saw in the corner Angela from my Brit Lit class drinking with some friends And now Angela and I had been... |
| | **UniCoRN**: ... **and** the McG first **and** *wearing* best **and** the **Man** ... the **story Man** ... **drinking** to the thing ... **and** I know ... |
| | **EEG-Text**: ... the boy climbs ... **with two** long arms ... **and two** ... pounces on **him and** tries **to** ... |
| | **BP-GPT**: ... **and** he's **like** loo**king** at **me** ... give **him** ... he *sees* this girl *coming* after ... that looks **like** a ... on the ground ... |
| | **Ours**: ... **Pie Man So I** was ... from **the Ram And** sure ... *Sheila Beale* student ... And now *Sheila* was different **from me and all** the other Fordham *students* ... *Sheila* ... was the kind of *student* ... |
| 40TR | **Target** : ... I called him Pie Man as he fled ... And I gave ... said that he cried out Ego sum non an bestia ... and he says Pie Man I love it ... |
| | **UniCoRN**: ... and **the scene** a platform **and I** and the rumor ... **stories** a *Sherila* ... non my place ... **my story** |
| | **EEG-Text**: ... **I'm** looking at her **and I** ... kind of **figure** as ... than *his* **and** then **it** ... got **good** educations ... went **to** work ... |
| | **BP-GPT**: ... know how you do **it** ... end of a long **day** ... *Clara* is hold**ing** on**to** my **hand** ... **The** gray haired **man** ... |
| | **Ours**: ... you **Pie Man And I** ... **And** wasn't I really **Pie Man** ... made *his* delivery and **fled** away *crying* **Ego sum non an bestia** ... what it **said** in ... |
| 60TR | **Target** : ... moved to New York ... And I went and looked ... what he did in Brazil And he said ... he wanted Bob to go hire ... because Alan was going to do this stupid thing ... And so I'm J Jhon Moscow ... |
| | **UniCoRN**: ... **he's** the girl You't **want** ... *Sherlock* so *Watson* ... two **things** with not .. Well *Arthur* ... almost days when before ... |
| | **EEG-Text**: ... **with** the press and **came to** *Boston* ... **And he said** ...he asked *Ryan* to find the same ... So *Ryan* contacted four officials and ... |
| | **BP-GPT**: ... **went** downtown **and** started ... **he knew** people **in the** group ... he helped **move** items but ... It **was** unclear who **was** ... from *Clara* ... |
| | **Ours**: ... **I did** not ... **I'm going to** *start* ... who **I am And so** ... **he** got plenty of ... But the fact that *Sheila* had collaborated with the *Dean* to ... radio station **in** *St Louis* ... |

As shown in Table 10 and 11, our proposed method, **CogReader**, outperforms SOTA methods in terms of capturing semantics and syntax in tokens, with more accuracy of individual words, which indicated in bold. Specifically, our decoding results capture more key content words ranging from verbs (such as "fled" and "cried") to nouns ("scene" and "night"), including more accurate named entities such as person and place names ("Pie Man" and "St Louis"), and produce sentences that are semantically more aligned with the intended meaning. Despite other SOTA methods are still able to decode some accurate information such as "Man" and "came", they fail to decode more meaningful words such as "The Ram" and "Ego sum non an bestia", which is paired in our method. Therefore, our method produces decoded outputs that exhibit higher word-level overlap and better semantic alignment with the reference texts, demonstrating superior decoding performance.

Table 11: Cases analysis for fMRI2text. Exact matches between the target and predicted sentences are indicated in **bold**, while semantic similarity is shown in *italic* font.

| fMRI Length | Results |
|---|---|
| 20TR | **Target**: ...I gave the figure ... I called him Pie Man ... as he fled ... And I gave ... said that he cried out Ego sum non an bestia ... |
| | **UniCoRN**: ... **and** *me* gray to little **the Man** and ..of years ... to the then **said** a case ... girl of the story ... |
| | **EEG-Text**: ... **I'm** trying to ... a *title* for the *book* ... stories by ... in *Waukegan* but he **was** ... **which** is about ... **So I am** going to ... |
| | **BP-GPT**: ... **and I** teach people *how* to **make** ... one day **I was** walk**ing** ... and she **said** ... you sound like ... |
| | **Ours**: ... **I** think ... happened next **Pie Man** ... **and fled** away *crying* **Ego sum non an bestia** Or ... **said** in my story... |
| 40TR | **Target** : ... she and her friends ... all the way ... at night .. She drives around ... I ever saw my mother cry ... My mom and dad ... was a government worker ... |
| | **UniCoRN**: ... **she** remembered how they used to ... the *evenings* sometimes ... when **she drove** back ... **and** that **was** when **she** felt ... |
| | **EEG-Text**: ... walking **down the** street **and** ... high**way** And then ... home from **work she** says to me ... There's something wrong with ... |
| | **BP-GPT**: ... about to **come off** ... **coming down** from the top **of** structure ... for no *good* reason ... playing with this guy ... |
| | **Ours**: ... **My mom** as **she cried** ... **she** stopped **and she drove** us home ... later that **night** .. An *organization* one of many ... onto **the** high**way** ... |
| 60TR | **Target** : ... he uh like moves ... Um the scene then ... the lady and she uh she looks ... come in Merlin like ... and leaves Um so ... |
| | **UniCoRN**: ... I says at takes and **then** of ... and um says ... him guy **like** ... **the scene** of ... she is the man of the friending to ... |
| | **EEG-Text**: ... **he** slowly walks ... **and moves** ... under **the** books she ... where something powerful had ... **he** tries to *walk away* ... |
| | **BP-GPT**: ... **uh he** stands rather ... when **she**'s about to open ... **and then so** ... when *Arthur* yells ... **and he** finally responds ... |
| | **Ours**: ... **like he** hears ... this is **uh** a **scene** takes ... Probably **um she looks like the** *woman* who ... **So** when *Mutarelli* **and** I ... |