# OpenReview forum: "Brain-Inspired fMRI-to-Text Decoding via Incremental and Wrap-Up Language Modeling"
_NeurIPS.cc/2025/Conference — NeurIPS 2025 spotlight_

### Official Review · Reviewer_aaPP · 2025-06-08

**Clarity:** 3
**Significance:** 2
**Originality:** 3
**Rating:** 4
**Confidence:** 2

**Summary:**

The paper introduces a new method for fMRI-to-text decoding, where they first pretrain a brain encoder on a large-scale fMRI dataset with a self-supervised MAE-style objective, then finetune the model on the target fMRI-to-text paired dataset. In particular, they introduce two mechanisms inspired by human language processing: segmental and incremental processing.

**Questions:**

Questions:
1. fMRI-to-text is like translation in NLP between two languages, where transformers succeed without explicitly adding segmental or incremental processing mechanisms. I am curious to understand why the authors think these will specifically benefit the problem of fMRI-to-text. Perhaps adding some elaboration would be helpful.
2. For Table 1, it seems that 60 > 40 > 20 TRs. I would be interested to see if increasing the number of TRs beyond 60 would lead to higher BLEU, ROUGE, and BERTScore?
3. What does the decoded text actually look like? Beyond the BLEU/ROUGH/etc. scores, it would be helpful to show some examples. For example, showing the original text, and different generations of decoded text under various random seeds. Also, perhaps it would be useful to qualitatively contrast this against generations from prior methods, so that it would be easier to understand the ways in which the proposed method makes an advance over prior methods.

**Ethical Concerns:**

["NO or VERY MINOR ethics concerns only"]

**Final Justification:**

The authors added new experimental results and motivations for their method. One weakness that is outstanding currently is that the qualitative results of the model still seem significantly different from the actual text (see my comment to the authors for more details). This makes it hard to tell if the method has actually made a substantial methodological advance toward solving the problem. Because of this, it is hard to evaluate the results in the context of the problem being tackled.

**Limitations:**

Yes

**Quality:**

3

**Strengths And Weaknesses:**

Strengths:
1. Good conceptual contributions of the text-guided masking strategy for MAE fMRI pretraining, and incremental, segmented fMRI-to-text decoding.
2. Their proposed method outperforms prior methods (UniCoRN, EEG-Text, BP-GPT) on all quantitative metrics evaluated (BLEU, ROUGE, BERTScore).

Weaknesses:
1. I think the paper would benefit from more qualitative results (see Question 3 below) to understand how the proposed innovations improve fMRI decoding, and how close their method is to "solving" the fMRI decoding problem.
2. I am interested in understanding the impact of their proposed sequential decoding framework (incremental and segmental processing) as that seems to be a key difference between this method and simply using a standard transformer architecture. However, this is challenging to understand empirically without a direct ablation. Referring to Table 2, first row, they ablate it when pretraining and text-guided masking are absent. But it would be most ideal, if possible, if there were an ablation where pretraining and text-guided masking are present, but not the sequential decoding strategy. This might be an important ablation to understand if adding structure (e.g., through their proposed sequential decoding method) is still helpful and important when there is sufficiently large pretraining data.

---

> ### Author Rebuttal · Authors · 2025-07-31
>
> We appreciate your valuable feedback, and our detailed responses are provided below.
>
> (1) W1&Q3 Examples of Qualitative Results：
> Due to page limitations, we provide the qualitative analysis results (i.e., actual text decoding examples) in the supplementary materials, specifically in Section A.3 (“Text Cases”) of the appendix. In this section, we present representative decoding examples of our method and SOTA approaches under different input lengths. The results show that our approach outperforms SOTA methods in capturing both semantic and syntactic information at the token level. It can identify more key content words, such as verbs, nouns, and named entities like person and place names. The generated sentences are also more semantically aligned with the reference texts. In the final version, we will include some of the decoded text in the main body of the paper.
>
> (2) W2 Ablation of Sequential Decoding：
> Following your suggestion, we added an ablation study to evaluate the effectiveness of Sequential Decoding. Specifically, we retained the text-guided masking module and the pretraining module, but replaced our proposed Sequential Decoding (i.e., segmented processing combined with incremental decoding and the wrap-up mechanism) with a holistic decoding approach. The new results, shown in the 4th row of the table below, demonstrate a clear performance drop when Sequential Decoding is removed. For example, BLEU-1 decreases from 38.3 to 21.6, ROUGE-R drops from 36.5 to 25.0, and BERTScore-R decreases from 55.1 to 47.2. These findings highlight that Sequential Decoding plays a crucial role in improving the decoding performance, validating its importance in our proposed framework. This ablation result will be added in our final version.
>
> |Sequential Decoding|Pretraining|Text-guided Masking|BLEU-1|BLEU-2|BLEU-3|BLEU-4|ROUGE-F|ROUGE-P|ROUGE-R|BERTScore-F|BERTScore-P|BERTScore-R|
> |-|-|-|-|-|--|-|-|-|-|-|-|-|
> |✗|✗|✗|17.7|6.5|2.4|1.1|29.2|32.9|24.7|46.7|47.6|45|
> |✓|✗|✗|32.5|16.5|11.1|8.9|28.2|25.8|30.6|51.1|50.3|51.8|
> |✓|✓|✗|35.9|20.5 |15.1|12.8|34|34.4|34.4|53.4|53.1|54.1|
> |✗|✓|✓|21.6|7.9|3.2|1.5|26.6|29.4|25|47.4|47.7|47.2|
> |✓|✓|✓|**38.3**|**23.4**|**18.3**|**15.9**|**36.4**|**37.6**|**36.2**|**54.4**|**53.8**|**55.1**|
>
> (3) Q1 Motivation of Segmental and Incremental Mechanisms based Method：
> fMRI-to-text decoding is fundamentally different from traditional machine translation tasks. Unlike translation, which performs a one-to-one mapping between two independent modalities, fMRI-to-text decoding reconstructs textual content from neural activity patterns generated during language comprehension of the human brain. As a result, models designed for cross-modal translation are not directly applicable to this task. Based on this insight, we hypothesize that adopting a decoding framework more aligned with human language processing mechanisms could better address this challenge. To this end, we propose a brain-inspired sequential decoding paradigm that incorporates both incremental processing and segmental integration. Both our experimental results (Section 4.3, Table 1) and ablation studies (Section 4.4, Table 2) strongly support this hypothesis. Compared with existing SOTA methods, our approach shows significant overall improvements, with particularly strong performance on long-sequence decoding tasks. The ablation studies further demonstrate that Sequential Decoding (i.e., segmented processing combined with incremental decoding and the wrap-up mechanism) contributes the most to performance gains, highlighting the importance and effectiveness of incorporating cognitively inspired components in fMRI-to-text decoding. In the final version, we will add a clearer explanation of the motivation behind segmental and incremental mechanisms.
>
> (4) Q2 More Results on Longer fMRI：
> Due to the limited availability of fMRI-text paired data, increasing the fMRI sequence length further cannot provide a sufficient number of training samples to meet the requirements for effective model training. For example, when the fMRI length is increased to 80 TRs, the total number of samples is only around 1,600. Therefore, in our main text, we report results only for 20, 40, and 60 TRs. Following the reviewer’s suggestion, we have also included decoding results for fMRI signal with 80TRs. We observed performance degradation across all metrics. However, where the drop is caused by the model’s limitation in handling longer sequences or by the lack of sufficient training data still needs further exploration. This question can be further investigated in future work when more data becomes available.
>
> |Length|BLEU-1|BLEU-2|BLEU-3|BLEU-4|ROUGE-F|ROUGE-P|ROUGE-R|BERTScore-F|BERTScore-P|BERTScore-R|
> |-|-|-|-|-|-|-|-|-|-|-|
> |20|25.4|10.5|4.7|2.6|23.4|22.6|24.6|46.3|45.7|46.9|
> |40|33.6|18.4|13.4|11.2|30.6|30.5|31.5|51|50.4|51.6|
> |60|38.3|23.4|18.3|15.9|36.4|37.6|36.2|54.4|53.8|55.1|
> |80|32.1|13.7|7.4|4.9|26.9|28|26.5|51|49.9|52.2|

---

> > ### Comment · Reviewer_aaPP · 2025-08-04
> >
> > (1) Metrics-wise, yes, I can see the numbers are higher. Qualitatively, like in Tables 4 and 5 of the Appendix, the result produced by your method (Ours) seems to be so different from the Target text. For example, the characters and verbs mentioned in the text are different, which renders the whole text to have a completely different meaning. By itself (not compared to the Target), it is also challenging to tell if the produced text is grammatically and semantically coherent. This is partially because you have provided snippets of the text with ellipses, rather than the full text. I understand that fMRI to text is challenging, but the qualitative results make it hard to tell if your method has actually made a substantial methodological advance toward solving the problem. Because of this, it is hard to evaluate the results in the context of the problem being tackled. This is not to take away from the strengths of the paper. I think the motivation and metrics improvement are good.
> >
> > (2) Thanks, this ablation is very helpful and worth including in the paper.
> >
> > (3) Thanks, this motivation is useful and would be good to include in the paper.
> >
> > (4) Thanks, this additional result on 80 TRs is helpful. What is the number of samples for different TRs? Does this use a sliding window, where increasing the TRs from 60 to 80 would just mean you have 20 fewer TRs at the end?

---

> > > ### Author Response · Authors · 2025-08-06
> > >
> > > We sincerely thank the reviewer for the additional feedback. We are pleased to hear that your concerns regarding our research motivation and experimental design have been addressed, and we appreciate your positive recognition of our work.
> > >
> > > Below, we respond to the new comments raised in your most recent review:
> > > 1. Regarding the quality of generated text:
> > > Thanks for your suggestion. We have included a comparative analysis of full decoded results from an example to more clearly demonstrate the quality of text generated by different methods. As shown in the text case below, our method consistently outperforms others in both semantic information extraction and narrative consistency. For instance, our model accurately identifies and generates key phrases such as “Pie Man”, “Ego sum non an bestia”, “scene”, “said” and “fled”, while producing more coherent and contextually appropriate sentences. In contrast, other methods exhibit greater deviations from the target content. More full decoded examples will be included in the supplementary material of the final version.
> > >
> > > |Text Case|
> > > |-|
> > > |**Target**: was **I** gave **the** figure a name I called him **Pie Man** capital P capital M **And** **I** described **him** as a cape wearing masked avenger Though in fact he’d been capeless And **said** **that** as he **fled** **the** **scene** he clicked his heels in rakish glee **And** **I** gave **him** a catchphrase in Latin I **said** **that** he **cried** out **Ego sum non an bestia** which means **I** am not an animal Which **makes** no sense|
> > > |**UniCoRN**: and me gray to the little **Man** and no taller than my knee, with a coat too big for his frame—lingered by the gate. It’s been nigh of years he to the then **said** a case here. The leather sat weathered on the step, its lock rusted shut, but I knew what was inside: the girl of the story, the one who scribbled tales of dragons in the margins of her notebooks, had tucked her most precious drawings there before she|
> > > |**EEG-Text**: and this is where it gets really interesting I’m trying to think of a title for the book and I finally come up with an idea that sounds like something out of one of those old pulp stories by John Wilmot who lived in Waukegan Illinois but he was actually born in Westchester County New York which is about twenty five miles outside Chicago So I am going to describe my novel as **The** **Man** Who Walks into Times Square and stops before he can get too lost|
> > > |**BP-GPT**: I’m a teacher and I teach people how to make things better by using my own talents And so one day I was walking down the street and this old lady came up to me and she **said** You know you sound like an intelligent black woman So I told her that's exactly what I'm here to do|
> > > |**Ours**: **I** think **that** you realize what happened next **Pie Man** emerged from **the** late night library drop **made** his delivery **and** **fled** away **crying** **Ego sum non an bestia** Or  that’s what it **said** in my story in the newspaper next day **which** ran with photos of **him** leaving **the** **scene** cape flowing behind **him** doing this **And** **I’m** just like praying my life doesn’t flash before my eyes **and** ruins|
> > >
> > > Additionally, as the reviewer pointed out, fMRI-to-text decoding remains a highly challenging task due to the substantial cross-modal gap between brain signals and natural language, as well as the complexity of the underlying mapping between the two domains. Another key difficulty lies in the relatively low temporal resolution of fMRI, where each TR corresponds to multiple words, making it harder to generate coherent and complete sentences. In future work, integrating fMRI with EEG could be a promising direction, as fMRI offers semantic representation capabilities, while EEG provides high temporal resolution. The complementary strengths of these modalities may help improve decoding accuracy. We will include this discussion in the limitation section of the final version.
> > >
> > > 2. Clarification on the research motivation:
> > > Thank you for recognizing the motivation behind our use of segmental and incremental mechanisms. We will incorporate this motivation more explicitly in the method section of the final version to better explain the cognitive inspiration and rationale behind our model design.
> > >
> > > 3. Sample counts and sliding window strategy for different TR lengths:
> > > On the Narrative dataset, the number of samples corresponding to different TR lengths is as follows: 20TR (10,621 samples), 40TR (5,267 samples), 60TR (3,486 samples), and 80TR (1,672 samples). To ensure semantic independence between segments, we did not apply a sliding window strategy. Instead, we used a non-overlapping segmentation approach with a fixed stride. This clarification will be included in the final version.

---

### Official Review · Reviewer_4tPt · 2025-07-01

**Clarity:** 3
**Significance:** 3
**Originality:** 3
**Rating:** 5
**Confidence:** 4

**Summary:**

This paper proposes CogReader, a brain-inspired framework for fMRI-to-text decoding that mimics human cognitive strategies for language comprehension. The key innovation is a sequential decoding approach that divides long fMRI sequences into segments, performs incremental decoding on each segment, and uses a wrap-up mechanism to maintain semantic continuity. Additionally, the authors introduce a text-guided masking strategy for fMRI representation learning that leverages BERT attention scores to identify semantically important timepoints.

**Questions:**

1. Can you provide evaluation on additional datasets to support broader applicability? Even preliminary results would strengthen the paper significantly.
2. What happens with dynamic segment length based on text complexity rather than fixed length? How do you analyze the trade-off between local and global context?
3. Have you experimented with more sophisticated semantic integration beyond BERT+MLP? How sensitive is performance to the wrap-up mechanism design?
4. Can you provide theoretical analysis of why segmented processing should outperform holistic approaches?
5. How does the method perform when text-guided masking isn't available during training?

**Ethical Concerns:**

["NO or VERY MINOR ethics concerns only"]

**Final Justification:**

My questions have been satisfactorily answered, and I find the overall contribution of the paper to be both valuable and suitable for acceptance at NeurIPS. I am raising my score from 4 to 5.

**Limitations:**

The authors adequately address some limitations in the discussion, particularly the static segmentation strategy and single dataset evaluation. However, they could better discuss: (1) potential negative effects of losing global context through segmentation, (2) computational overhead of the sequential approach, and (3) applicability constraints when paired training data isn't available.

**Quality:**

3

**Strengths And Weaknesses:**

Strengths:
1. The framework meaningfully incorporates human language processing mechanisms into computational design, which is well-motivated and represents a principled approach.
2. The experimental design is comprehensive with proper baselines, thorough ablation studies, and validation against noise data.

Weaknesses:
1. The method is only evaluated on one dataset, which significantly limits generalizability claims.
2. While cognitively inspired, the paper lacks deeper theoretical analysis of why segmented processing should outperform holistic approaches.
3. The wrap-up mechanism (BERT + MLP) and attention score aggregation seem overly simplistic given the complexity involved.

---

> ### Author Rebuttal · Authors · 2025-07-31
>
> We appreciate your valuable feedback, and our detailed responses are provided below.
>
> (1) W1&Q1 Evaluation on Other Dataset：
> We further evaluated our method on an additional public dataset (LeBel et al., 2023), which includes fMRI recordings from 8 participants when they listened to 27 full-length natural narrative stories (totaling about 6 hours). The experimental configuration follows the same setup as used for the Narratives dataset in the study. The results for the 60TR decoding task are presented in the table below. As shown, our method also consistently achieves superior performance compared to current SOTA methods. These results will be added to the supplementary materials of our final version.
>
> |Methods|BLEU-1|BLEU-2|BLEU-3|BLEU-4|ROUGE-F|ROUGE-P|ROUGE-R|BERTScore-F|BERTScore-P|BERTScore-R|
> |-|-|-|-|-|-|-|-|-|-|-|
> |UniCoRN|21.5|5.4| 2.3|0.6|17.1|14.1|19.7|40.1|37.7|42.7|
> |EEG-Text|23.6|7.5| 2.2|1.0|27.6|28.1|26.1|47.2|47.3|47.2|
> |BP-GPT|20.7|8.1| 2.6|1.2|27.1|30.4|23.5|50.7|51.3|49.2|
> |CogReader(ours)|**35.2**|**18.0**|**10.2**|**7.1**|**32.2**|**34.0**|**30.9**|**53.9**|**54.5**|**53.3**|
>
> [1] LeBel et al., 2023. A natural language fMRI dataset for voxelwise encoding models. Scientific Data, 10(1): 555.
>
> (2) W2&Q4 Theoretical analysis of why segmented processing outperform holistic approaches：
> Segmented processing offers clear theoretical advantages over holistic approaches, which can be explained from both cognitive science and modeling perspectives.
> From a neuroscience perspective, our segmented and sequential decoding framework that integrates incremental decoding and the wrap-up mechanism is consistent with numerous findings on how the human brain processes language. Prior studies have shown that humans typically divide linguistic input into smaller, manageable units (e.g., clauses or sentences) and integrate information at natural boundaries (Tiffin-Richards et al., 2018). This strategy effectively reduces cognitive load and improves comprehension accuracy. Behavioral evidence further indicates that when text passages are approximately 55 words long, humans achieve optimal reading efficiency and comprehension at both normal and fast reading speeds (Dyson et al., 2001). The optimal segment length identified in our model, 20 TRs (about 60 words), closely aligns with these cognitive findings, providing strong theoretical support for our brain-inspired segmented decoding design.
> From a modeling perspective, fMRI signals inherently have a low signal-to-noise ratio, and holistic decoding of long time series significantly increases the difficulty of extracting meaningful information. Inspired by the advantages of state-passing approaches for handling long sequences, we divide the fMRI time series into multiple segments, perform incremental decoding on each segment, and use the wrap-up mechanism to pass a semantic summary of the previous segment to guide the decoding of the next. This design not only mitigates performance degradation caused by excessive sequence length but also improves semantic consistency across segments.
> Comparative experiments with existing holistic decoding approaches (e.g., UniCoRN, EEG-Text, BP-GPT) demonstrate that our segmented decoding method achieves significantly better performance, further validating its effectiveness and practical advantages.
>
> [1] Tiffin-Richards et al., 2018. The development of wrap-up processes in text reading: A study of children’s eye movements. Journal of Experimental Psychology: Learning, Memory, and Cognition, 44(7): 1051.
> [2] Dyson et al., 2001. The influence of reading speed and line length on the effectiveness of reading from screen. International Journal of Human-Computer Studies, 54(4): 585-612.
>
> (3) W3&Q3 Over-Simplicity and Sensitivity of the Wrap-up Mechanism：
> Our proposed wrap-up mechanism is designed to semantically compress the textual information decoded from the previous fMRI segment and pass it as a summary vector to guide the decoding of the next segment. Thus, it is not a simple state propagation but rather a cross-modal semantic integration process involving modality translation and higher-level abstraction. Specifically, the mechanism first decodes an fMRI segment into text, encodes the generated text using BERT to obtain a rich semantic representation, and then projects this representation into the fMRI latent space of the subsequent segment via an MLP to assist in decoding. This design achieves semantic summarization and cross-modal transfer in a lightweight manner, avoiding the introduction of excessive parameters that could lead to unstable training.
> We systematically evaluated different wrap-up designs and found that more complex models do not necessarily yield better performance. For example, in experiments with varying MLP hidden dimensions, a dimension of 128 achieved the best performance on both word-level and semantic-level metrics, whereas increasing the dimension to 256 led to a significant drop in performance, even falling below that of 32 or 64 dimensions (Section 4.2, Figure 5). We also explored more sophisticated frameworks, such as integrating BLIP for semantic aggregation (Li et al., 2022). However, experimental results showed that the BLIP-based approach achieved comparable performance to the MLP-based approach but exhibited less stable training, likely due to its higher data requirements. Based on these findings, we ultimately adopted the simpler and more stable BERT+MLP design. In future work, we plan to explore more efficient wrap-up mechanisms to achieve better semantic compression and further improve model performance.
>
> [1] Li et al., 2022. Blip: Bootstrapping language-image pre-training for unified vision-language understanding and generation. In International conference on machine learning 2022 Jun 28 (pp. 12888-12900).
>
> (4) Q2 Fixed Segmentation scheme：
> The primary goal of this study was to validate the feasibility of the proposed brain-inspired fMRI-to-text decoding framework, particularly to examine whether the incremental decoding and wrap-up mechanisms can improve decoding performance for long text sequences. Therefore, in the current work, we adopted a relatively simple segmentation strategy with a fixed segment length. Through systematic experiments, we verified that using a segment length of 20 TRs (approximately 60 words) yields the best decoding performance. Interestingly, this result is consistent with findings from cognitive science showing that humans achieve optimal comprehension when reading passages of around 55 words (Dyson et al., 2001). This consistency provides theoretical support for the fixed segmentation strategy adopted in our study.
> As discussed in our paper, we acknowledge that a fixed segmentation strategy may split the narrative at unnatural points, resulting in improper semantic segmentation. Determining how to segment text effectively is a challenging but important topic. In future work, we plan to explore content-adaptive segmentation methods that dynamically predict segment boundaries based on narrative complexity or syntactic structure, enabling the model to flexibly adapt to diverse textual inputs. Furthermore, we aim to investigate approaches that combine dynamic segmentation with more flexible cross-segment information integration to achieve a better balance between local decoding accuracy and global semantic coherence.
>
> [1] Dyson et al., 2001. The influence of reading speed and line length on the effectiveness of reading from screen. International Journal of Human-Computer Studies, 54(4): 585-612.
>
>
> (5) Q5 Ablation of text-guided masking：
> In Section 4.4, we conducted an ablation study to verify the effectiveness of text-guided masking. Specifically, we retained the pretraining module (HCP dataset pretraining in the first stage) and the sequential decoding module (i.e., incremental and wrap-up based fMRI-to-Text decoding framework) while removing the text-guided masking strategy, and then evaluated the model’s performance (see the 3th row in the table below). To enhance the semantic relevance of learned fMRI representations, the proposed text-guided masking strategy identifies words with high semantic relevance, then prioritizes the reconstruction of fMRI time points associated with these key words, which directs the model’s attention toward fMRI signals associated with highly informative semantic content, thereby improving the semantic relevance and effectiveness of fMRI representations. The results show that removing text-guided masking led to a decrease in decoding performance across all metrics, further demonstrating the importance of this module in improving the model’s performance.
>
> |Sequential Decoding|Pretraining|Text-guided Masking|BLEU-1|BLEU-2|BLEU-3|BLEU-4|ROUGE-F|ROUGE-P|ROUGE-R|BERTScore-F|BERTScore-P|BERTScore-R|
> |-|-|-|-|-|--|-|-|-|-|-|-|-|
> |✗|✗|✗|17.7|6.5|2.4|1.1|29.2|32.9|24.7|46.7|47.6|45|
> |✓|✗|✗|32.5|16.5|11.1|8.9|28.2|25.8|30.6|51.1|50.3|51.8|
> |✓|✓|✗|35.9|20.5 |15.1|12.8|34|34.4|34.4|53.4|53.1|54.1|
> |✓|✓|✓|**38.3**|**23.4**|**18.3**|**15.9**|**36.4**|**37.6**|**36.2**|**54.4**|**53.8**|**55.1**|

---

> > ### Comment · Reviewer_4tPt · 2025-08-06
> >
> > Thank you to the authors for addressing my comments. My questions have been satisfactorily answered, and I find the overall contribution of the paper to be both valuable and suitable for acceptance at NeurIPS. Please ensure that the additional results are included in the revised version. I am raising my score from 4 to 5.

---

> > > ### Author Response · Authors · 2025-08-06
> > >
> > > Thanks for your positive feedback and for taking the time to review our paper. We sincerely appreciate your recognition of our contributions and are glad to hear that your concerns have been addressed. We will make sure to include all additional results and clarifications discussed in the final version of the paper. Thank you again for your thoughtful evaluation and support.

---

> ### Author Response · Authors · 2025-08-05
>
> Dear Reviewer 4tPt,
>
> We sincerely appreciate the time and effort you have devoted to reviewing our submission. We have carefully prepared our rebuttal, aiming to address the valuable points you raised in a direct and respectful manner.
>
> Should you have an opportunity to review our response, we would be grateful for any additional thoughts or clarifications you may wish to share. Thank you once again for your thoughtful and constructive feedback.
>
> Best regards, The Authors

---

### Official Review · Reviewer_nUQu · 2025-07-02

**Clarity:** 2
**Significance:** 3
**Originality:** 3
**Rating:** 4
**Confidence:** 4

**Summary:**

The paper proposes a method for decoding fMRI to text using a segmented approach inspired by how humans process language to mange the load in the working memory. Instead of decoding the full fMRI sequence at once, the input is split into segments that are decoded incrementally. After each segment, there is a “wrap-up” phase that integrates the current semantic understanding and passes it forward to inform the next decoding step, helping maintain coherence across segments. In addition to this sequential decoding setup, the authors introduce a two-stage pretraining scheme for learning better fMRI representations. First, they use a large-scale dataset for general pretraining. Then, they propose a text-guided masked autoencoder (MAE) that leverages attention scores from the associated text to guide which fMRI time points to mask and reconstruct—essentially forcing the model to focus on semantically important parts. Experiments on the Narratives dataset show that their approach outperforms existing methods, especially for longer sequences. In addition, the paper shows ablation studies to support the design decisions including the advantage of using sequential decoding, pretraining + text-guided pretraining, studies on the segment length, and various other experiments like testing the model with random data.

**Questions:**

See weakness, but here are the most important questions:
1. How are the splits done for the text-guided pretraining and decoding. Looking briefly at the supplementary material I can see the splits done at 60, 20, 20, however, can you please confirm whether the exact same splits are used for both stages (to prevent pretraining on data that is used for evaluating the decoding process).
2. Can you clarify why in UniCoRN they report 34% in BLEU score, whereas in this paper the number is different.

A minor thing:
3. In line 296, instead of "time series of 20 TRs", was it meant to be written "time series of 60 TRs"?

**Ethical Concerns:**

["NO or VERY MINOR ethics concerns only"]

**Final Justification:**

I thank the authors for their thoughtful rebuttal and additional experiments. While the changes in the rebuttal address all my main concerns, some overall limitations remain in the work. In particular, the reliance on a fixed window size and the assumption in having access to paired fMRI-text data, may not hold in many real-world applications. Overall, their experiments clearly show that their method improves decoding performance (fmri-to-text) in their evaluation datasets. Therefore, I am increasing my score from 3 to 4.

**Limitations:**

The paper mentions already some critical limitations like the fixed segment size, as well as evaluation in a single public dataset. However, in addition, it is important to mention that even the text-guided pretraining is strictly speaking supervised and as such it is limited in applying at other datasets.

**Paper Formatting Concerns:**

In the checklist, "Experiment statistical significance" is marked as [NA], with the justification "see Experiments." However, this is not truly not applicable. Even if the training is computationally expensive and large-scale significance testing is impractical, the lack of statistical analysis should be acknowledged as a limitation—not marked as [NA].

**Quality:**

3

**Strengths And Weaknesses:**

##  Strengths:

1. Text-guided masking pretraining: The paper introduces a novel text-guided masking strategy during pretraining. By using attention scores from the text to guide which fMRI timepoints to reconstruct, the model learns to focus on semantically important tokens. This is an intersting idea that ties textual semantics directly to fMRI representation learning.
2. Sequential, segment-based decoding: Inspired by human language processing, the model decodes fMRI inputs in segments, with a wrap-up phase that integrates prior context. This helps preserve semantic flow across longer sequences and is a step forward compared to one-shot decoding approaches.
3. Thorough experimental validation: The authors present extensive ablation studies and comparisons against existing methods. Their approach consistently outperforms baselines, especially for longer fMRI sequences.

## Weaknesses:

1. Limited contextual integration in wrap-up phase: While the paper claims to model human-like wrap-up effects, the implementation appears to simply pass the sentence embedding of the previous segment (through an MLP) into the decoder for the next segment. This means the model only conditions on the immediately preceding segment, not the full history, which weakens the claim of human-like contextual integration. The current explanation of the wrap-up phase is a bit vague and it would be valuable to clarify this mechanism and test a version where more cumulative context is aggregated. For example, it's unclear how the wrap-up embedding is merely summed with the fMRI embedding (the only information available is the symbol ⊕ symbol in the figure, but it is nowhere described whether those embeddings are simply added).
2. Potentially unfair comparison to prior work: The paper shows that performance of prior models degrades with longer sequences, but this comparison may be unfair. Earlier methods (e.g., UniCoRN) are not designed to decode \textit{long} sequences at once, and in the original paper it is presented by decoding signals of a few TRs (in fact looking at UniCoRN paper they provide a table showing that the length T=10 works best for their experimental setting). In other words, if UniCoRN works better by first segmenting, and then decoding each segment independently, than that is a better baseline instead of using UniCoRN with a very long sequence which might suffer due to the big context window. Additionally, some strong results from prior work—such as BLEU scores around 34.77% reported in UniCoRN—are not directly reconciled with the numbers presented here. It is important to use the setup for which UniCoRN is designed for and compare to this approach.
3. Text-guided masking is not self-supervised: Although the pretraining approach is compelling, it depends on access to paired text to compute attention scores. This limits its applicability as a general-purpose pretraining strategy. Moreover, this stage is performed on the same dataset used for evaluation, raising concerns about generalization and potential data leakage. It is not clearly stated whether the same splits were used across pretraining and decoding to avoid this issue. If not carefully separated, this could significantly inflate performance.
4. Fixed segmentation scheme: Segmenting the fMRI time series into fixed-length chunks may lead to cuts at unnatural points in the narrative, disrupting the underlying semantic coherence. Exploring adaptive or linguistically informed segmentation (e.g., based on sentence or clause boundaries) could improve results.
5. Unclear role of wrap-up in actual fMRI decoding: It's unclear whether the observed performance gain from the wrap-up phase stems from better modeling of brain activity or simply from smoother language generation. As an ablation, it would be useful to decode segments independently (as in UniCoRN), and then apply a language model post-hoc to enforce consistency, isolating the contribution of the fMRI model from the decoding pipeline.
6. The model is only evaluated at a single public dataset. Although fMRI-to-Text are difficult datasets to get, there are still other publically available datasets where one can validate the approach.

---

> ### Author Rebuttal · Authors · 2025-07-31
>
> We appreciate your positive comments on our work, including “novel strategy” and “interesting idea”. Detailed responses to all comments are provided below.
>
> (1) W1 Limited contextual integration in wrap-up：
> Our wrap-up mechanism performs incremental semantic integration rather than relying solely on the immediately preceding segment. At each step, the text decoded from the current fMRI segment is encoded using BERT to obtain a rich semantic representation, which is then projected into the fMRI feature space of the next segment via an MLP to guide subsequent decoding. Because this process is applied recursively, the semantic embedding passed to the next segment already incorporates information from all previous segments, accumulated through the progressively decoded text. In other words, the wrap-up embedding is not merely the representation of a single sentence but a continuously integrated semantic summary that reflects the entire preceding context. We will provide a more detailed explanation of this process in the final version.
>
> (2) W2&Q2 Unfair comparison to prior work：
> Following the reviewer’s suggestion, we modified UniCoRN’s decoding strategy for 60TR sequences by dividing them into 10TR segments, decoding each segment independently, and concatenating the outputs. As shown in the table below, our method still substantially outperforms UniCoRN. We attribute this superiority to two key innovations: 1) the introduction of a text-guided masking strategy during representation learning, which improves the semantic relevance and effectiveness of the learned fMRI representations; and 2) the use of a sequence-wise decoding framework with a wrap-up mechanism that leverages semantic information from preceding segments to guide subsequent decoding, thereby improving accuracy while maintaining semantic coherence in the generated text.
>
> |Method|BLEU-1|BLEU-2|BLEU-3|BLEU-4|ROUGE-F|ROUGE-P|ROUGE-R|BERTScore-F|BERTScore-P|BERTScore-R|
> |-|-|-|-|-|-|-|-|-|-|-|
> |UniCoRN|23.2|7.4|3.3|1.6|21.8| 23.9|17.5|46.2| 46.7| 46.7|
> |CogReader(ours)|**38.3**|**23.4**|**18.3**|**15.9**|**36.4**|**37.6**|**36.2**|**54.4**|**53.8**| **55.1**|
>
> Additionally, the discrepancy between the results reported for UniCoRN and ours stems from a data leakage issue during its evaluation phase, which has also been reported in other paper (Yin et al., 2024). Specifically, according to the code provided by UniCoRN, its decoder model uses the same forward method for both training and inference. During inference, the decoder_input_ids (shifted ground-truth labels) are fed into the model, allowing real text to influence the generation process and thus causing data leakage. In our evaluation, we fixed this issue by removing the ground-truth input during inference. As a result, the BLEU scores decreased compared to those originally reported for UniCoRN.
>
> [1] Yin et al., 2024. Language reconstruction with brain predictive coding from fmri data. arXiv preprint arXiv:2405.11597. 2024.
>
> (3) W3&Q1 Text-guided masking：
> Our fMRI representation learning adopts a “self-supervised pretraining + text-guided fine-tuning” approach. Specifically, we first perform self-supervised pretraining on the HCP dataset (non-fMRI-text paired data) and then apply fine-tuning on fMRI-text paired data using the text masking strategy proposed in this work. The adoption of this strategy has two advantages: 1) it alleviates the limitation of scarce fMRI-text paired data; 2) it effectively guides the model to focus on fMRI time points corresponding to high-semantic-value text tokens, thereby enhancing the semantic relevance and effectiveness of the learned representations. The comparison with other representation learning methods (see Section 4.5, Table 3) validates the effectiveness and advantages of our method.
> Regarding the potential issue of data leakage, we clarify that the first-stage fMRI representation learning and the second-stage decoding adopt an identical data-splitting scheme. Specifically, the data are split based on stimuli into training (60%), validation (20%), and test (20%) sets, which are saved in separate directories. During both the representation learning and decoding stages, the model loads data exclusively from these corresponding directories, ensuring that the data splits used in both stages are completely consistent and that there is no overlap between the data used for pretraining and that used for decoding evaluation. Therefore, the results reported in this paper are free from any risk of data leakage.
>
> (4) W4 Fixed Segmentation Scheme：
> The primary goal of this study was to validate the feasibility of the proposed brain-inspired fMRI-to-text decoding framework, particularly to examine whether the incremental decoding and wrap-up mechanisms can enhance decoding performance for long text sequences. Therefore, our focus was on evaluating the overall framework performance rather than extensively exploring segmentation strategies. To this end, we adopted a relatively simple yet effective fixed segmentation scheme and experimentally verified that a segment length of 20 TRs (approximately 60 words) achieves the best decoding performance. This finding is consistent with results from cognitive science, which show that humans achieve efficient reading and optimal comprehension with text passages of about 55 words (Dyson et al., 2001). Such consistency provides theoretical support for our choice of 20 TRs as the optimal segment length.
> However, as noted in both the discussion section and the reviewers’ comment, a fixed segmentation strategy has inherent limitations because it may split the narrative at unnatural points. In future work, we plan to explore content-adaptive segmentation methods that dynamically predict segment boundaries based on narrative complexity or syntactic structure, allowing the model to more flexibly adapt to diverse textual inputs.
>
> [1] Dyson et al., 2001. The influence of reading speed and line length on the effectiveness of reading from screen. International Journal of Human-Computer Studies, 54(4): 585-612.
>
> (5) W5 Unclear role of wrap-up in actual fMRI decoding：
> We conducted an additional experiment to further validate the effectiveness of the wrap-up mechanism. Specifically, we divided the 60TR fMRI sequence into three segments, performed independent decoding for each segment, and then applied GPT-2 to assess the semantic coherence between consecutive segments, and generate simple connecting phrases and adjust words in original segments to enhance consistency. As shown in the table below, our proposed wrap-up-based method achieves significantly better decoding performance compared to this post-processing approach. We will include the results of this ablation study in the final version.
>
> |Method|BLEU-1|BLEU-2|BLEU-3|BLEU-4|ROUGE-F|ROUGE-P|ROUGE-R|BERTScore-F|BERTScore-P|BERTScore-R|
> |-|-|-|-|-|-|-|-|-|-|-|
> |post_hoc|26.9|9.7|4.5|2.1|28.3|30.6|26.1|49.3|48.4|50.5|
> |Wrap-up|**38.3**|**23.4**|**18.3**|**15.9**|**36.4**|**37.6**|**36.2**|**54.4**|**53.8**|**55.1**|
>
> (6) W6 Evaluation on Other Dataset：
> We conducted additional comparative experiments on a new public dataset (LeBel et al., 2023). This dataset contains fMRI signals recorded from 8 participants when they listened to 27 complete and natural narrative stories (with a total duration of approximately 6 hours). The experimental setup is consistent with that used for the Narratives dataset in the paper, and the comparison results for 60TR fMRI signals are shown in the table below. As can be seen, our method still significantly outperforms existing SOTA methods. These results will be included in the supplementary materials of our final version.
>
> |Methods|BLEU-1|BLEU-2|BLEU-3|BLEU-4|ROUGE-F|ROUGE-P|ROUGE-R|BERTScore-F|BERTScore-P|BERTScore-R|
> |-|-|-|-|-|-|-|-|-|-|-|
> |UniCoRN|21.5|5.4| 2.3|0.6|17.1|14.1|19.7|40.1|37.7|42.7|
> |EEG-Text|23.6|7.5| 2.2|1.0|27.6|28.1|26.1|47.2|47.3|47.2|
> |BP-GPT|20.7|8.1| 2.6|1.2|27.1|30.4|23.5|50.7|51.3|49.2|
> |CogReader(ours)|**35.2**|**18.0**|**10.2**|**7.1**|**32.2**|**34.0**|**30.9**|**53.9**|**54.5**|**53.3**|
>
> [1] LeBel et al., 2023. A natural language fMRI dataset for voxelwise encoding models. Scientific Data, 10(1): 555.
>
> (7) Q3 Writing Error：
> Thank you for pointing out this writing error. We will correct it in our final vision.

---

> ### Comment · Reviewer_nUQu · 2025-08-04
> **Response**
>
> Thank you for your rebuttal and addressing my comments. Below the acknowledgement and my comments:
>   1) Thank you for clarifying that the wrap-up embedding is updated recursively through the decoding process. This helped address my concern about the limited context integration.
>   2) I appreciate the effort in re-evaluating UniCoRN under a segmented setup and correcting for potential data leakage during inference. I do suggest, however, including comparisons to this work as well (Yin et al. (2024)), which you cite in your rebuttal but do not include in the main experiments. Since that work is also motivated by human language processing, a direct comparison would help position your contribution more clearly within the existing literature. Not sure why this work was not included in the paper in the first place.
>   3) While I understand the distinction between self-supervised pretraining and text-guided fine-tuning, I would still consider the second stage to be supervised, as it requires access to text annotations. This does not diminish the novelty of the masking strategy, but it is important to be clear about its limitations.
>   4) Leaving the dynamic segmentation as future work. In addition, I looked at other reviewers’ comments and the authors’ rebuttals. I agree that using the fixed setup is a natural starting point; however, I do not agree with the authors drawing strong parallels to how humans process information (e.g., rebuttal to Reviewer 4tPt and W2/Q4 mentioning "strong theoretical support" or even in the title "Human-Like Comprehension"). One can phrase that the segmented approach is **inspired** by how humans process information; however, I think presenting the work as building a model "like the human brain" goes way beyond what the work is about. There are many differences, for example, the use of pretraining, the wrap-up phase (where an fMRI segment is decoded into text, passed through BERT, and then the BERT representation is projected back into the fMRI latent space via an MLP to guide the next decoding step). These are all quite different from how the human brain would plausibly process information. Claiming something this strong would require much deeper investigation and a range of analyses to support it. The work itself remains firmly on the application side, namely focused on advancing state of the art in fMRI-to-text decoding. Referencing findings like "using 60 words is consistent with 55-word reading chunks that support efficient comprehension" (Dyson et al., 2001) is an interesting observation, but it does not come close to justifying a claim that the method is decoding language like the human brain. The architecture is entirely different, and there is no analysis connecting model behavior to actual brain activity.
>      - To be clear, I see value in the paper, especially in its contributions to improving decoding performance. But extending that into claims about brain-like processing is simply not supported. In fact, thinking about it now, it feels excessive to have that statement even in the title. Until reading the rebuttal, I had treated the cognitive framing as light motivational context and **not** as a core part of the novelty.
>
> 5 & 6) Including the additional experiments for better understanding on the role of wrap-up phase, and generalization of the method to new datasets.
>
> I am willing to potentially increase my score by 1 in light of the additional experiments and clarifications provided. However, it’s important to be cautious not to overclaim novelty by drawing strong parallels to human processing. In my view, this work does not offer any explanation or insight into how the brain actually processes language, nor does it establish a meaningful connection between the proposed model and human cognitive function. The parallels drawn, such as the 55- vs. 60-word segmentation, are, at best, curious observations that would require much deeper investigation to support any stronger interpretation. Ultimately, this is an application-focused paper: it presents a well-designed method and shows how computational modeling can push the state of the art in fMRI-to-text decoding. In my opinion, that is where its strength lies, not in modeling or explaining brain function.

---

> > ### Author Response · Authors · 2025-08-07
> >
> > We sincerely thank the reviewer for taking the time to engage in detailed discussions. We are glad to have addressed your concerns regarding the wrap-up embedding and appreciate your acceptance of our fixed-length segmentation strategy. We also value your positive feedback on the additional experiments and the practical innovation of our work.
> >
> > Below we respond to the new points you raised:
> >
> > (1) Comparison with other method:
> > Thanks for your suggestion. We initially did not include Yin et al.’s method in our comparison because, during the writing phase, we contacted the authors via email but didn’t obtain the code and model. They informed us that an updated version was in development and suggested waiting for its release before conducting comparisons, with no follow-up before the submission deadline.
> > Following your advice, we implemented the available version of PREDFT for updated comparison experiments. As shown in the table below, PREDFT achieves slightly better decoding performance than the other three baseline methods. This may be attributed to its incorporation of a brain-inspired predictive coding mechanism, which anticipate upcoming content based on prior context during reading, thereby enhancing semantic relevance in generation. Despite this improvement, the overall performance of PREDFT still falls short compared to our method, especially for the long text decoding. Upon analysis, the main reason may be lie in its use of a conventional holistic decoding strategy. Given fMRI’s low SNR, holistic processing of long temporal sequences hinders extracting meaningful semantic information. In contrast, inspired by human language comprehension, we employed a segmented decoding framework that divides the fMRI time series into segments, performs incremental decoding on each segment, and uses a wrap-up mechanism to pass a semantic summary from the previous segment to guide the next. This design not only mitigates performance degradation caused by long sequences but also improves semantic consistency across segments. The corresponding experimental results and analysis will be included in the final version.
> >
> > |Length|Method|BLEU-1|BLEU-2|BLEU-3|BLEU-4|ROUGE-F|ROUGE-P|ROUGE-R|BERTScore-F|BERTScore-P|BERTScore-R|
> > |-|-|-|-|-|-|-|-|-|-|-|-|
> > |20TR|UniCoRN|22.9|2.5|0.3|0|20.3|19.6|21|43.9|44.2|42.8|
> > ||EEG-Text|24.6|9.3|4.4|1.9|21.9|21.1|23.4|44.6|43.9|45.4|
> > ||BP-GPT|21.6|3.8|2.5|1.7|21.6|20.9|23.4|44.1|42.1|46.3|
> > ||PREDFT|24.3|4.2|0.7|0.1|20.1|22.3|18.3|45.9|45.5|46.7|
> > ||CogReader(ours)|**25.4**|**10.5**|**4.7**|**2.6**|**23.4**|**22.6**|**24.6**|**46.3**|**45.7**|**46.9**|
> > |40TR|UniCoRN|19.1|2.3|0.5|0.1|17.8|18.2|17.6|43.8|44.8|42|
> > ||EEG-Text|20.1|7.3|3|1.3|24.4|25.1|24.7|45.4|45.8|45.5|
> > ||BP-GPT|19.9|3.6|2.3|1.5|21.1|19.3|22.9|42.6|39.4|46.1|
> > ||PREDFT|25.9|4.8|1.4|0.4|21.1|24.8|18.6|46.3|46.2|46.8|
> > ||CogReader(ours)|**33.6**|**18.4**|**13.4**|**11.2**|**30.6**|**30.5**|**31.5**|**51**|**50.4**|**51.6**|
> > |60TR|UniCoRN|18|1.7|0.2|0.4|16.5|15.9|17|43.2|43.7|42.7|
> > ||EEG-Text|22.1|8.2|3.4|1.6|28.1|29.4|28.1|47.7|47.8|47.7|
> > ||BP-GPT|19.3|3.4|1.3|0.6|19.4|19.6|19.3|41.6|38.2|45.3|
> > ||PREDFT|26.4|6.1|1.9|0.6|28.1|25.5|20.5|48.1|47.7|48.5|
> > ||CogReader(ours)|**38.3**|**23.4**|**18.3**|**15.9**|**36.4**|**37.6**|**36.2**|**54.4**|**53.8**|**55.1**|
> >
> > (2) Clarification on supervised learning:
> > We agree with your point regarding the second-stage text-guided fine-tuning should be categorized as supervised learning due to its reliance on annotated data. We will revise the “limitations” section in the final version to acknowledge the supervised nature of this stage.
> >
> > (3) On the use of "human-like" claims:
> > We fully acknowledge your concerns about the use of the term "human-like" and agree that our current model is better described as brain-inspired, drawing from principles observed in human language comprehension rather than attempting to replicate the mechanisms of human cognition. While our results provide preliminary support for the feasibility of incorporating cognitively inspired mechanisms such as incremental decoding and semantic wrap-up, we acknowledge that these do not constitute modeling actual brain function.
> > Accordingly, we will revise the title of the paper to: "Brain-Inspired fMRI-to-Text Decoding via Incremental and Wrap-Up Language Modeling" in our final version. Furthermore, we will revise all the paper to ensure that all related descriptions throughout the paper are updated to reflect a brain-inspired framing rather than human-like based.

---

> > > ### Comment · Reviewer_nUQu · 2025-08-08
> > > **Response**
> > >
> > > Thank you for addressing my concerns. I will increase my score from 3 to 4. Please make sure to update the paper with the new changes.

---

> > > > ### Author Response · Authors · 2025-08-08
> > > >
> > > > We sincerely appreciate your positive feedback and the increase in score. We will ensure that all additional results and the newly discussed revisions are incorporated into the final version of the paper.

---

### Official Review · Reviewer_4jo4 · 2025-07-06

**Clarity:** 3
**Significance:** 2
**Originality:** 3
**Rating:** 4
**Confidence:** 5

**Summary:**

This paper proposes "CogReader," a new framework for decoding natural language text from fMRI signals, specifically designed to handle long sequences. The authors identify that existing methods, which process an entire fMRI sequence at once, suffer from performance degradation on longer inputs. To address this, their method is inspired by human language comprehension and involves two main contributions. First, it employs a sequential decoding strategy where the fMRI time series is divided into fixed-length segments. Each segment is decoded incrementally, and a "wrap-up" mechanism summarizes the output and passes it as a semantic prior to the next segment to maintain continuity. Second, for learning fMRI representations, the paper introduces a text-guided masking strategy within a Masked Autoencoder (MAE) framework, which aims to focus the model on semantically important time points. Experiments on the Narratives dataset show that the proposed method outperforms state-of-the-art approaches, with the performance gap widening as the length of the decoded text increases

**Questions:**

1 The central framing of the paper is its "human-like" approach. However, the core mechanism is a fixed-length segmentation of the fMRI signal. Could you provide a stronger justification from cognitive science literature that supports fixed-length (as opposed to content-driven, adaptive) segmentation as a plausible model for human language comprehension?

2 Given that the overall architecture (HCP pre-trained encoder + LLM decoder) is quite standard, could you more precisely delineate the novelty of your "wrap-up" mechanism compared to other methods that handle long sequences or context, such as hierarchical models or models with state-passing mechanisms? Clarifying this might help in assessing the originality of the contribution.

3 The absence of qualitative examples makes it difficult to fully evaluate your model's output. Could you please provide a few representative examples of decoded text from your model and the baseline models for a 60 TR sequence? A discussion of the comparative error types (e.g., semantic drift, repetition, hallucination) would be valuable

**Ethical Concerns:**

["NO or VERY MINOR ethics concerns only"]

**Final Justification:**

Thanks for the rebuttal. If the authors can waive in all their described modifications in their revised manuscript, I can raise my recommendation to boarderline accept.

**Limitations:**

Yes

**Quality:**

3

**Strengths And Weaknesses:**

Weaknesses

1. The paper's claim to novelty is somewhat overstated. The general framework—pre-training an fMRI encoder on a large dataset like HCP , then fine-tuning it to produce embeddings that are fed into a conditional generation model or LLM (like BART )—is a standard paradigm in both visual and language decoding from fMRI. This approach has been widely used in prior work such as [1-3] which are not cited, including the cited UniCoRN paper.  This means the primary novelty rests almost entirely on the incremental decoding and "wrap-up" mechanism. While this is a reasonable contribution, it feels more like an incremental improvement on a common backbone rather than a fundamentally new framework.


2.  The framing of the work as "Human-Like Language Comprehension" is not sufficiently justified. While inspired by cognitive science, the implementation uses fixed-length segmentation (N_s=20 TRs), which is more of an engineering heuristic to manage memory than a true model of human cognition. Human reading involves dynamic, content-aware segmentation, not arbitrary fixed-length chunking. The paper lacks a deeper discussion or evidence to support why this specific mechanism is a faithful model of human comprehension.


3. A major weakness in evluation is the complete absence of qualitative examples of the decoded text in the main paper or appendix (the appendix is mentioned but not provided). Metrics like BLEU and ROUGE are known to be limited and can be misleading. Without seeing the samples actual generated sentences, it is diffcult to assess crucial aspects like semantic coherence, factual accuracy, and the specific types of errors the model makes compared to baselines.


Missed references include but are not limited to:

[1] Seeing beyond the brain: Conditional diffusion model with sparse masked modeling for vision decoding. CVPR 2023

[2] Contrast, Attend and Diffuse to Decode High-Resolution Images from Brain Activities. NeurIPS 2023

[3] MapGuide: A Simple yet Effective Method to Reconstruct Continuous Language from Brain Activities. NAACL 2024



Strengths

1 The paper addresses a well-defined and significant problem in the field of neural decoding. The difficulty of decoding long, continuous language sequences from fMRI is a major bottleneck, and the authors propose a clear strategy to mitigate this issue.

2 The quantitative results presented are strong. Table 1, in particular, makes a compelling case for the method's effectiveness, showing that while baseline performance degrades with longer sequences (from 20 TRs to 60 TRs), the proposed CogReader model shows marked improvement. The ablation studies also systematically validate the contribution of each component.


3  The paper is generally well-written and structured. The core ideas are explained clearly, and the figures, especially Figure 2, provide a helpful illustration of the overall architecture and data flow

---

> ### Author Rebuttal · Authors · 2025-07-31
>
> We sincerely appreciate the reviewers’ positive feedback on our research topic, experimental results, and overall presentation of the paper. Below, we provide detailed responses to all comments.
>
> (1) W1 Novelty of Work：
> Our work is very novel on achieving better brain signal decoding, in particular, addressing or mitigating the insufficiency of data and long-context challenges.
> First, we propose a brain-inspired sequential decoding paradigm for fMRI-to-text decoding that incorporates both incremental processing and segmental integration. The motivation for this framework arises from the fundamental differences between fMRI-to-text decoding and conventional cross-modal translation tasks. Unlike translation, which performs a one-to-one mapping between two independent modalities, fMRI-to-text decoding reconstructs textual content from neural activity patterns generated during language comprehension. As a result, models designed for cross-modal translation are not directly applicable to this task. Based on this insight, we hypothesize that adopting a decoding framework more aligned with human language processing mechanisms could better address this challenge. Our experimental results (Section 4.3, Table 1) and ablation studies (Section 4.4, Table 2) support this hypothesis, demonstrating the feasibility of brain-inspired fMRI-to-text decoding and providing a new perspective for future model design. In the final version, we will add a clearer explanation of the motivation behind this brain-inspired framework.
> Second, we design a text-guided masking strategy during fMRI representation learning to guide the model to focus on fMRI time points that correspond to high-value text tokens, thereby enhancing the semantic relevance and expressiveness of the learned fMRI representations. The comparison with other representation learning methods (Section 4.5, Table 3) further demonstrates the effectiveness and advantages of this approach.
>
> (2) W2&Q1 Fixed Segmentation Scheme：
> The fixed segment length of 20 TRs was determined via systematic validation in our work (Section 4.2, Figure 4). This finding is roughly consistent with evidence from cognitive science, which shows that text passages of approximately 55 words support efficient reading at both normal and fast speeds and lead to the highest comprehension levels (Dyson et al., 2001). Given that each TR (1.5s) corresponds to roughly three words, a 20TR segment covers approximately 60 words, aligning well with the optimal text length for comprehension reported in. This provides theoretical support for selecting 20 TRs as the optimal text segment length in our work.
> Since the main objective of this study was to validate the feasibility of applying brain-inspired mechanism to fMRI-to-text decoding, our focus was on evaluating the overall framework performance rather than extensively exploring text segmentation strategies. As discussed in our paper and highlighted by the reviewers, our future work will further investigate content-adaptive segmentation methods that dynamically predict segment boundaries based on narrative complexity, enabling the model to more flexibly adapt to diverse textual inputs.
>
> [1] Dyson et al., 2001. The influence of reading speed and line length on the effectiveness of reading from screen. International Journal of Human-Computer Studies, 54(4): 585-612.
>
> (3) W3&Q3 Examples of Decoded Text：
> Due to page limitations, detailed examples of decoded text are provided in the supplementary materials. Specifically, in Section A.3 ("Text Cases") of the Appendix, we provide representative decoded examples for our method and SOTA approaches across different input lengths. Our proposed method outperforms SOTA methods in terms of capturing semantics and syntax in tokens and captures more key content words ranging from verbs to nouns, including more accurate named entities such as person and place names, and produce sentences that are semantically more aligned with the intended meaning. Therefore, our method produces decoded outputs that exhibit higher word-level overlap and better semantic alignment with the reference texts, demonstrating superior decoding performance. In the final version, we will include some of the decoded text in the main body of the paper.
>
> (4) Q2 Novelty of wrap-up mechanism：
> In our framework, the proposed wrap-up mechanism is superficially similar to existing state-passing mechanisms but differs fundamentally in both the modality and the structural flow of the transferred information.
> Traditional state-passing mechanisms typically operate within a single modality (e.g., text-to-text), propagating hidden states from one segment to the next to maintain temporal continuity, akin to how recurrent neural networks or autoregressive models carry over internal states across time steps. In contrast, our wrap-up mechanism is a cross-modal semantic summarization process: it decodes an fMRI segment into text, encodes the generated text using BERT to extract a rich semantic representation, and then projects this representation back into the fMRI latent space of the subsequent segment via an MLP to guide future decoding. This is not merely state propagation, but a semantic-level integration that involves modality translation and higher-level abstraction.
> While our method shares superficial resemblance to state-passing, particularly in its sequential setup, it is structurally distinct. State-passing emphasizes retention and transfer of temporal internal states (e.g., hidden activations) across steps, whereas our wrap-up mechanism summarizes each fMRI window into a semantic embedding that represents integrated content, without explicitly transferring low-level hidden states. In that sense, it is more analogous to a block-wise summarization approach, such as using a [CLS] token or attention-based pooling to encode the essence of an input segment.
> We believe this difference is not just architectural but also functional: the wrap-up mechanism aims to provide a semantic bridge across fMRI segments by leveraging language-domain priors, rather than preserving low-level continuity.

---

> ### Author Response · Authors · 2025-08-05
>
> Dear Reviewer 4jo4,
>
> We sincerely appreciate the time and effort you have devoted to reviewing our submission. We have carefully prepared our rebuttal, aiming to address the valuable points you raised in a direct and respectful manner.
>
> Should you have an opportunity to review our response, we would be grateful for any additional thoughts or clarifications you may wish to share. Thank you once again for your thoughtful and constructive feedback.
>
> Best regards,
> The Authors

---

### Note · Authors · 2025-08-12

We thank the AC and reviewers for their constructive discussion, and especially appreciate Reviewers #nUQu, #4tPt, and #aaPP for actively engaging in the discussion, providing positive feedback on our rebuttal, and raising their scores. We highlight three key points:

(1)**Novelty and Motivation**: We clarified the fundamental differences between fMRI-to-text decoding and conventional translation tasks, and explained the motivation and novelty of our brain-inspired sequential decoding framework, which integrates incremental decoding with a wrap-up mechanism. Experimental and ablation results show that this framework delivers markedly better decoding performance than existing holistic strategies, especially for long text sequences. We also detailed the design rationale of the wrap-up mechanism and confirmed its role in improving both semantic consistency and decoding accuracy through additional ablation studies.

(2)**Model Performance**: On a newly tested dataset, our method consistently outperforms current state-of-the-art approaches, demonstrating both robustness and strong generalization capability across datasets.

(3)**Fixed Segmentation Strategy**: Through systematic validation, we determined that a segment length of 20 TRs (approximately 60 words) delivers optimal performance. This finding is consistent with cognitive science research, which indicates that passages of about 55 words enable efficient reading at both normal and fast speeds and achieve the highest comprehension levels, thus providing theoretical support for our current choice. Furthermore, as noted in our Discussion, we also recognize that fixed segmentation may introduce unnatural semantic boundaries and plan to explore content-adaptive segmentation methods in our future work.

All clarifications and additional results will be incorporated into the camera-ready.

---

### Decision · Program_Chairs · 2025-09-17

**Decision:**

Accept (spotlight)

**Comment:**

This paper introduces a framework for decoding natural language text from fMRI signals that is specifically designed to handle long sequences. Current methods process entire fMRI sequences at once and experience performance degradation with longer inputs. To address this issue, the proposed method involves two main contributions. First, it uses a sequential decoding strategy in which the fMRI time series is divided into fixed-length segments. Each segment is decoded incrementally, and a "wrap-up" mechanism summarizes the output and passes it as a semantic prior to the next segment to maintain continuity. Second, the paper introduces a text-guided masking strategy within a masked autoencoder (MAE) framework for learning fMRI representations. This strategy focuses on semantically important time points. Experiments on the Narratives dataset demonstrate that the proposed method outperforms SOTA approaches, with a wider performance gap as the length of the decoded text increases.
The paper is interesting and convincing but makes slight overclaims. These have been identified and discussed with the reviewers and will be mitigated in the final version. Additionally, experiments will be presented on a second dataset, which is a significant advantage.
Overall, the authors did a good job of presenting and defending their strategy, and the performance gap with respect to SOTA is large enough to warrant publication. While the paper's message is not extremely sophisticated, it is certainly useful.